# Towards Whole-corpus Reconstruction of Heterogeneous RAG Knowledge Bases

Peiru Yang [* 1 2]   Yi Luo [* 1]   Zhenfeng Gao [3]   Tong Ju [4]   Haoran Zheng [1]   Linjie Zhu [3]   Hongke Fu [3]   Qing Li [1]
Shangguang Wang [1]   Tao Qi [1]

## Abstract

Retrieval-Augmented Generation (RAG) systems are increasingly deployed to provide query-based access to large knowledge bases, thereby introducing concrete privacy risks whereby the underlying corpus may be partially or fully extracted through the deployed service. Existing extraction attacks typically rely on locally driven search strategies, in which newly extracted content is inferred or expanded based on previously recovered fragments. However, real-world knowledge bases are often multi-source and heterogeneous, with pronounced semantic discontinuities across domains. Such gaps can trap extraction methods that rely on local semantic continuity in local optima, severely limiting large-scale corpus reconstruction. In this paper, we introduce an extraction framework (GeoEx) designed to navigate and reconstruct heterogeneous RAG knowledge bases without any prior knowledge. The framework plans extraction directly in the embedding space of a proxy retrieval model to improve global coverage, and employs an embedding inversion module to convert latent vectors into executable queries. We further propose a composite geometric strategy that combines orthogonal query synthesis for cross-domain exploration with local embedding perturbations for dense extraction within discovered clusters. Experiments on mixed corpora spanning eight diverse domains and multiple retrievers and LLMs show that GeoEx significantly outperforms baselines in both extraction coverage and query efficiency.

---

[*]Equal contribution [1]School of Computer Science, Beijing University of Posts and Telecommunications [2]Department of Electronic Engineering, Tsinghua University [3]Sangfor Technologies Inc., Shenzhen, China [4]Northwestern Polytechnical University. Correspondence to: Tao Qi <taoqi.qt@gmail.com>.

*Proceedings of the 43rd International Conference on Machine Learning*, Seoul, South Korea. PMLR 306, 2026. Copyright 2026 by the author(s).

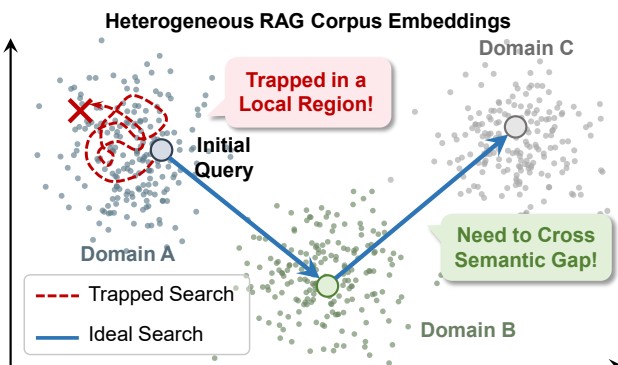

**Heterogeneous RAG Corpus Embeddings**

*Figure 1.* Illustration of the search navigation challenge in data extraction attacks towards heterogeneous RAG corpus.

## 1. Introduction

Retrieval-Augmented Generation (RAG) provides up-to-date, domain-specific information for LLMs by leveraging external knowledge (Lewis et al., 2020b; Gao et al., 2023; Huang et al., 2025). In real-world deployments, these external knowledge bases typically comprise heterogeneous corpora drawn from multiple domains, which are often highly valuable and sensitive (Zou et al., 2025; Zeng et al., 2024; Wang et al., 2025). However, this reliance on sensitive external corpora also introduces new security risks, as the underlying knowledge base may be exposed through the system's query interface, potentially leading to significant privacy breaches (Ni et al., 2025; Qi et al., 2025). A central threat in this setting is the data extraction attack against the RAG corpus, where attackers systematically query a RAG system to reconstruct its underlying knowledge base.

Most existing work on RAG extraction focuses on recovering individual facts or targeted fields, and evaluates attack success at the level of whether a specific sensitive record can be reconstructed (Cohen et al., 2024; Zeng et al., 2024). They typically rely on strong prior knowledge of the corpus content, such as specific keywords, topic distributions, or known text snippets, to craft effective initial queries. For example, Zeng et al. (2024) utilize specific semantic prefixes to target and extract personally identifiable information from the retrieval database. Intuitively, it is relatively straightforward to prompt an LLM to reproduce retrieved content for

a specific target when prior knowledge about that target is available. However, such priors are typically unavailable when attempting to scale this approach to reconstruct an entire knowledge base. In the absence of these hints, selecting queries that can efficiently cover an opaque corpus under a limited query budget becomes more challenging.

Recent studies have begun to investigate whole-corpus extraction under a no-prior setting, typically leveraging LLM-based agents to heuristically rewrite queries based on extracted topics or textual relevance inferred from retrieved contexts (Jiang et al., 2025; Di Maio et al., 2024). However, in practical deployments, a knowledge base often aggregates multiple corpora spanning diverse domains. In such heterogeneous settings, these heuristic strategies are prone to becoming trapped in local optima (Fig. 1), leading to limited corpus coverage and inefficient query utilization. Therefore, how to efficiently and effectively attack such heterogeneous knowledge bases is still an challenging problem, especially when the adversary has no knowledge of the corpus composition in the RAG systems.

Therefore, we introduce a RAG data extraction framework (GeoEx) for heterogeneous knowledge bases behind black-box RAG systems, assuming no prior knowledge. With an initial query as the entry point, GeoEx shifts the search planning from the discrete text domain to a continuous embedding space. In the embedding space, GeoEx can systematically plan the long-term exploration, and then project these planned vectors back into natural language queries to probe the target RAG system. This design allows us to move beyond local semantic clusters and achieve cross-domain transitions across multiple corpus distributions. Besides, since the RAG retriever in unknown, we employ a surrogate encoder to establish a proxy embedding space for exploration. We further introduce a coverage-driven composite geometric strategy to balance global navigation across distinct semantic domains with local densification within each discovered region, which can thereby evolve from minimal seed queries into a high-coverage reconstruction of the heterogeneous corpus. Experiments on mixed corpora constructed from eight diverse domains, evaluated across two retrievers and five SOTA LLMs, demonstrate that GeoEx significantly outperforms baselines in extraction coverage by 19.94% on average. **Code** is available at `https://github.com/ypr17/GeoEx`. Our contributions are as follows:

- We introduce GeoEx, a novel extraction framework designed to systematically reconstruct heterogeneous RAG knowledge bases in a no-prior setting.

- We develop a surrogate-based inverse navigation mechanism that employs a composite geometric strategy to translate latent plans into queries, balancing global exploration and local exploitation to maximize coverage.

- Extensive experiments on heterogeneous corpora across three retrievers and five LLMs demonstrate that GeoEx significantly outperforms strong baselines in both extraction coverage and query efficiency.

## 2. Related Works

Privacy-oriented attacks targeting RAG corpora can be broadly grouped into two categories. The first is Membership Inference Attacks (MIAs), which assess whether a particular data point exists in the retrieval database (Li et al., 2025; Liu et al., 2025; Yang et al., 2025). The second is data extraction attacks, which go a step further by directly recovering and revealing the contents of the corpus, resulting in explicit leakage of private information (Zeng et al., 2024; Cohen et al., 2024; Qi et al., 2025).

**RAG Membership Inference Attacks**. Anderson et al. (2024) propose the earliest MIA against RAG, which directly queries the system whether a candidate document is present in the context. Building on this, Li et al. (2025) assess membership by comparing generated outputs with candidate samples through semantic similarity, while Liu et al. (2025) further introduce masking-based perturbations and infer membership from the model's prediction robustness under such edits. Although MIA and data extraction both target privacy leakage from the retrieval corpus, they operate at fundamentally different levels. MIAs are limited to an existence-level, binary decision problem in which the adversary already holds a candidate data point and determines whether it is included in the corpus. By contrast, extraction attacks seek to recover full text from the corpus, revealing substantive content rather than membership alone. This leads to significantly greater privacy risk and more closely reflects the capabilities of real-world adversaries.

**RAG Data Extraction**. Some works explore data leakage from the RAG corpus by using prompt-injection strategies to extract targeted pieces of information from the retrieval corpus (Zeng et al., 2024; Cohen et al., 2024). For example, Zeng et al. (2024) introduce a composite structured prompting method that uses an information query plus a command to extract sensitive content from the RAG retrieval database. Cohen et al. (2024) propose a jailbreaking-based RAG extraction attack that uses embedding-collision queries to force retrieval of target documents from the RAG datastore. These approaches focus on recovering individual knowledge units or selectively eliciting specific information, relying on manually crafted anchors or commands. As a result, they are limited in scope and do not scale to full-corpus extraction. More recent work aims to recover entire RAG knowledge bases, demonstrating stronger attack capabilities and more severe privacy risks (Qi et al., 2025; Jiang et al., 2025). Qi et al. (2025) investigate prompt-injected RAG data extraction, showing that adversaries can exploit

instruction-following LLMs to elicit and reconstruct retrieved datastore content directly. Jiang et al. (2025) present an agent-based automated attack that iteratively extracts private corpus chunks by leveraging leaked text to generate new adversarial queries. Di Maio et al. (2024) propose an adaptive attack that iteratively explores the private datastore through relevance-guided query generation. However, these methods operate on a single, domain-homogeneous corpus, whereas real-world RAG deployments typically integrate multiple heterogeneous sub-knowledge bases spanning diverse domains. This mismatch limits the generality and robustness of existing extraction approaches.

## 3. Methods

### 3.1. Preliminaries

**Problem Formulation**. We consider a multi-domain RAG system whose external knowledge base is a union of multiple sub-corpora $\mathcal{KB} = \bigcup_{m=1}^{M} \mathcal{KB}^{(m)}$, where each $\mathcal{KB}^{(m)}$ is stored as a collection of chunks. An RAG extraction attack is an interactive process where an adversary issues a sequence of queries $[q_1, ..., q_B]$ to the system and observes the returned chunks, with a fixed query budget $B$. The goal of the adversary is whole-corpus reconstruction, that is, to maximize the coverage of the underlying chunk set $\mathcal{KB}$ by the recovered chunk set $\widehat{\mathcal{KB}}_B$ obtained within budget $B$.

**Threat Model**. We adopt a black-box threat model where the adversary can only interact with the RAG system through its query interface. We further assume a no-prior setting in which the adversary does not have any knowledge on the corpus $\mathcal{KB}$, and must adapt its strategy solely from the observed system outputs to improve extraction coverage.

### 3.2. Motivation

Efficient whole-corpus extraction necessitates a mechanism to systematically traverse the high-dimensional semantic space with controllable step sizes and specific directions. However, we observe a fundamental disconnect where manipulating discrete textual tokens often fails to produce predictable changes in the target embedding space. Minor modifications in the text can trigger disproportionate semantic shifts or result in negligible variations, making it impossible to precisely steer the search trajectory or regulate the exploration magnitude. To bridge this gap, we draw inspiration from recent advances in embedding inversion (Morris et al., 2023; Chen et al., 2024; 2025), which demonstrate that continuous latent representations can be effectively translated back into natural language. Motivated by this, we establish a surrogate continuous space for planning, enabling GeoEx to mathematically define precise search strategies before projecting them back into the discrete query format. To operationalize this, we utilize a publicly available encoder

to construct a fully observable proxy space, bypassing the inaccessibility of the black-box retriever. Crucially, our framework operates on the premise that the semantic topology is generally consistent across different state-of-the-art retrieval models. Consequently, high-value search directions identified in this surrogate space are likely to remain effective when transferred to the target system.

The latent distribution of a heterogeneous corpus is typically non-uniform, characterized by high-density domain clusters, such as medical records or legal contracts, separated by vast low-density void regions. Standard search strategies often become trapped in local optima within these isolated groups, as reliance on semantic similarity makes it difficult to break free from the current domain boundaries. From a geometric perspective, a direction orthogonal to the current search subspace represents the path of least correlation and maximum information gain. Therefore, orthogonal synthesis offers the optimal geometric solution to traverse semantic gaps and reach disjoint knowledge islands. Complementing this global discovery, we further require fine-grained sampling to exhaustively cover the interior of each identified cluster. By applying small-magnitude perturbations to valid retrieval hits, we can effectively perform dense mining of local regions to reconstruct specific knowledge fragments.

### 3.3. Latent Navigation via Surrogate Inversion

The primary obstacle in corpus reconstruction is the inaccessibility of the target retriever's internal parameters and vector representations. To overcome this opacity, we employ a publicly available encoder as a surrogate to construct a proxy embedding space. This fully observable space allows us to perform precise geometric calculations and planning that are impossible to define in the discrete text domain.

Building on this proxy representation, we model the extraction process as an automated *Retrieve-Plan-Invert* loop, summarized in Appendix A. Formally, let $\mathcal{E}_S(\cdot)$ denote the surrogate encoder and $\mathcal{M}$ be a pre-trained language model used for decoding. In each iteration, given a discovered context chunk $c$, we map it to the latent proxy space as $\mathbf{z}_c = \mathcal{E}_S(c)$. Next, a geometric planning function $\Phi$ operates on $\mathbf{z}_c$ to compute a target vector $\mathbf{v}^*$, designed to bridge semantic gaps or densify local neighborhoods. Since $\mathbf{v}^*$ lies in continuous latent space, we employ a similarity-guided inverse decoding mechanism to project it back into a discrete natural language query $q$. At each decoding step $t$, we adopt a re-ranking strategy to align the generation trajectory with the planned vector. Specifically, we first obtain a candidate set $\mathcal{C}_t$ comprising the top-$K$ tokens predicted by $\mathcal{M}$. For each candidate $w \in \mathcal{C}_t$, we append it to the existing context $w_{<t}$ and re-encode the complete sequence using the surrogate encoder $\mathcal{E}_S$ to obtain a lookahead embedding. We then compute a modified ranking score $S(w)$ by injecting

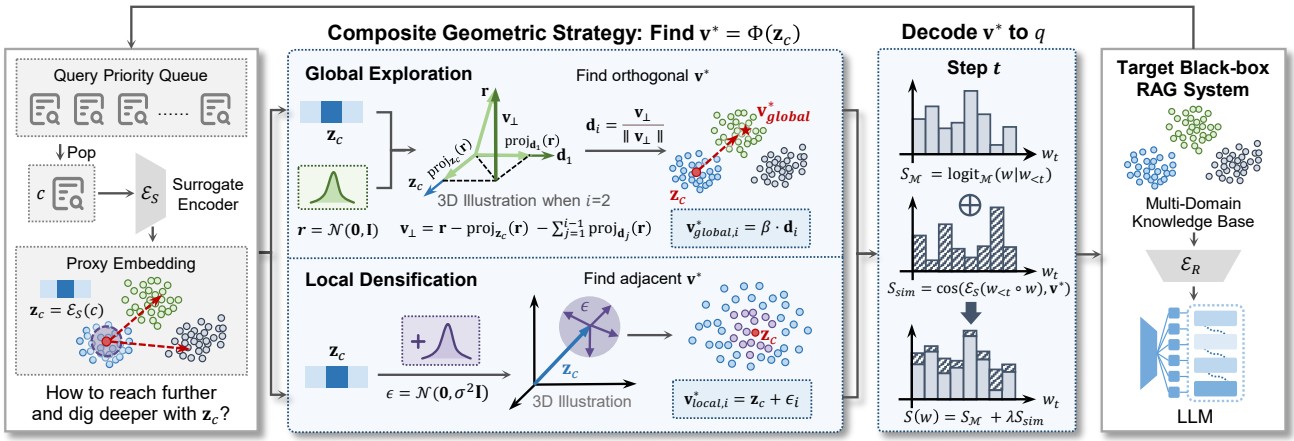

*Figure 2.* Overview of GeoEx. The framework performs geometric planning in a surrogate embedding space, utilizing both global exploration and local densification strategies, and then inversely decodes the planned vectors into textual queries.

the semantic similarity bias into the model's raw logits:

$$S(w) = \text{logit}_{\mathcal{M}}(w|w_{<t}) + \lambda \cdot \cos(\mathcal{E}_S(w_{<t} \circ w), \mathbf{v}^*), \quad (1)$$

where $\circ$ denotes the concatenation operation, and $\lambda$ is a hyperparameter controlling the strength of the semantic guidance. This decoding strategy forces the generated query $q$ to semantically approximate the planned trajectory $\mathbf{v}^*$ while maintaining the linguistic fluency ensured by $\mathcal{M}$. The synthesized query is then issued to the target RAG system to retrieve new evidence, thereby closing the navigation loop.

The overall execution of GeoEx follows a self-sustaining iterative protocol. Initialized with a minimal set of seed queries, the system maintains a priority queue of discovered knowledge chunks. In each cycle, a chunk is popped to serve as a semantic anchor in the surrogate space. The geometric planning module then generates target vectors to explore new regions, which are inverted into queries to probe the black-box RAG system. Newly retrieved chunks are added to the extraction set and fed back into the queue, enabling the framework to progressively expand its reconstruction coverage from isolated starting points to the broader heterogeneous corpus within a fixed query budget.

### 3.4. Coverage-Driven Composite Geometric Strategy

To effectively navigate the heterogeneous landscape of the target RAG corpus, GeoEx employs a composite geometric strategy that alternates between two distinct modes: *Global Exploration* for crossing domain boundaries, and *Local Densification* for mining neighborhood details. This design prevents the search process from getting trapped in high-density semantic clusters while ensuring thorough coverage of identified regions effectively.

**Global Exploration via Orthogonal Jumps**. The main challenge for extraction is overcoming directional bias that

traps the search in the current domain. Standard retrieval expansion methods, which rely on semantic similarity, tend to retrieve chunks that are increasingly redundant or confined within the same topic (e.g., retrieving more medical records when the goal is to find legal documents). To force a semantic transition, we need to generate queries that are maximally uncorrelated with the current context. Intuitively, to maximize the divergence from known information, we seek a trajectory that is geometrically orthogonal to the subspace spanned by the current retrieval context.

Formally, let $\mathbf{z}_c = \mathcal{E}_S(c)$ be the surrogate embedding of the currently retrieved context chunk. To maximize exploration diversity, we aim to generate a set of $N$ distinct target vectors that are orthogonal not only to the current context $\mathbf{z}_c$ but also to each other. We achieve this via an iterative Gram-Schmidt process. Let $\mathcal{D} = \{\mathbf{d}_1, \ldots, \mathbf{d}_{i-1}\}$ denote the set of normalized orthogonal directions generated in previous steps. In the $i$-th iteration, we first sample a random Gaussian vector $\mathbf{r} \sim \mathcal{N}(\mathbf{0}, \mathbf{I})$. We then compute the strictly orthogonal component $\mathbf{v}_\perp$ by removing the projections of $\mathbf{r}$ onto both the current context $\mathbf{z}_c$ and all previously discovered directions in $\mathcal{D}$:

$$\mathbf{v}_\perp = \mathbf{r} - \text{proj}_{\mathbf{z}_c}(\mathbf{r}) - \sum_{j=1}^{i-1} \text{proj}_{\mathbf{d}_j}(\mathbf{r}), \quad (2)$$

where $\text{proj}_{\mathbf{u}}(\mathbf{r}) = \frac{\mathbf{r}^\top \mathbf{u}}{\|\mathbf{u}\|^2} \mathbf{u}$. If $\|\mathbf{v}_\perp\|$ falls below a numerical stability threshold, $\mathbf{r}$ is discarded and re-sampled. The valid $\mathbf{v}_\perp$ is then normalized to obtain the unit direction $\mathbf{d}_i = \frac{\mathbf{v}_\perp}{\|\mathbf{v}_\perp\|}$, which is added to $\mathcal{D}$. Finally, we construct the $i$-th global target vector by scaling this direction:

$$\mathbf{v}^*_{global,i} = \beta \cdot \mathbf{d}_i. \quad (3)$$

This procedure is repeated until $N_g$ orthogonal targets are synthesized, ensuring the resulting queries explore struc-

turally disjoint regions of the semantic space. By decoding $\mathbf{v}_{global}^*$ using the inverse navigation mechanism (Eq. 1), GeoEx generates queries that are semantically valid but structurally disjoint from the current topic, effectively prompting the retriever to jump to a new corpus region.

**Local Densification via Gaussian Perturbation**. While orthogonal jumps effectively identify new semantic clusters, they do not guarantee exhaustive coverage of the information within those clusters. Once a valid chunk is retrieved from a new domain, it serves as a high-value anchor for fine-grained mining. To uncover the dense, granular knowledge surrounding this anchor, we shift our strategy from directional leaps to local isotropic sampling. Based on the premise that the retriever's latent space exhibits local smoothness, we induce controlled variations to the anchor embedding to probe its immediate neighborhood.

Formally, for a newly discovered context chunk $c$, we treat its surrogate embedding $\mathbf{z}_c = \mathcal{E}_S(c)$ as the centroid of a local semantic region. We generate a set of local target vectors by injecting isotropic Gaussian noise into this centroid:

$$\mathbf{v}_{local,i}^* = \mathbf{z}_c + \boldsymbol{\epsilon}_i, \quad \text{where } \boldsymbol{\epsilon}_i \sim \mathcal{N}(\mathbf{0}, \sigma^2 \mathbf{I}). \quad (4)$$

This sampling is performed $N_l$ times to produce local targets for local exploration. Here, $\sigma$ is a small scaling factor (relative to the global step size $\beta$) controlling the perturbation magnitude. Unlike the large orthogonal shifts intended to traverse semantic voids, this small-scale noise keeps the target vector $\mathbf{v}_{local}^*$ well within the "gravitational pull" of the current cluster. By decoding these perturbed vectors, GeoEx synthesizes diverse paraphrases and nuance-shifting queries that systematically sweep the local manifold, maximizing the extraction of nearby chunks.

# 4. Experiments

## 4.1. Experimental Setups

**RAG System Implementations**. We adopt five widely used SOTA LLMs as the generator component in our evaluated RAG systems, covering both proprietary and open-source families, including GPT-4o (Hurst et al., 2024), Qwen2.5-32b (Team et al., 2024), Deepseek-v3 (Liu et al., 2024), Gemini-2.5 (Comanici et al., 2025), and GPT-5 (Singh et al., 2025). We instantiate the retriever of the target RAG systems with Contriever (Izacard et al., 2021), BGE (Xiao et al., 2024), or E5 (Wang et al., 2022) to cover widely used embedding-based retrieval backbones.

**Surrogate Encoders and Inverse Model**. We employ GTR-T5 (Ni et al., 2022) as the surrogate encoder on the attacker side to induce a surrogate embedding space for query navigation and inverse decoding, without access to the target retriever representations. The BART (Lewis et al., 2020a) model is used as the inverse decoding language model.

**Hyperparameters**. Unless otherwise stated, all experiments adopt the following default configurations. The query budget is fixed at 2000, and each query retrieves top-$K = 2$ chunks from the target retriever. The seed query number is set to 1. For inverse decoding, we use decode top-$K = 256$ candidate tokens with inverse decoding length of 80. For global exploration, we synthesize 4 orthogonal targets per iteration. For local densification, Gaussian noise with small variance $\sigma$ is applied around newly discovered anchors. The domain number ranges from 4 to 8, with 4 as default.

**Knowledge Base Construction**. We build a multi-domain knowledge corpus by integrating eight widely used datasets spanning biomedical research, finance, scientific literature, law, patent documents, movie analysis, encyclopedic knowledge, and elementary science reasoning. Specifically, the corpus includes Bio (PubMedQA), Finance (Bloomberg), Arxiv (arXiv), Law (LexGLUE), Patent (BigPatent), Movie (IMDB), Wiki (Wikipedia), and Science (ARC). Together, these datasets simulate a heterogeneous multi-domain retrieval environment for evaluating beyond single-domain or homogeneous knowledge settings.

**Baselines**. We compare GeoEx with four representative RAG extraction baselines, including GAB (Zeng et al., 2024), DGEA (Cohen et al., 2024), PINE (Qi et al., 2025), and RAG-thief (Jiang et al., 2025). Since DGEA also relies on a surrogate encoder, we evaluate two variants under our setup: a same-source setting where the surrogate encoder matches the target retriever, denoted as the aligned variant (A), and a cross-source setting where the surrogate encoder differs from the target retriever, denoted as the misaligned variant (M), mirroring the variants considered for GeoEx.

## 4.2. Evaluation Protocols

We evaluate extraction quality using two complementary metrics, coverage and efficiency, computed on unique retrieved chunks after deduplication. Let $\mathcal{KB}$ denote the full chunk set in the target knowledge base, and $\mathcal{C}_B$ denote unique chunks retrieved after issuing $B$ queries with top-$K$ retrieval. Coverage measures corpus recovery: $\text{Coverage}(B) = \frac{|\mathcal{C}_B|}{|\mathcal{KB}|} \times 100\%$. Efficiency measures budget utilization: $\text{Efficiency}(B) = \frac{|\mathcal{C}_B|}{B \cdot K} \times 100\%$. Unless otherwise specified, we use $B = 2000$ when reporting coverage, $B = 100$ when reporting efficiency, and set $K = 2$ across all experiments for fair comparison. Intuitively, coverage represents the extent of corpus recovery under a relatively sufficient budget, while efficiency captures the slope of the initial rising trend during the early extraction phase.

## 4.3. Main Results

Table 1 summarizes the main results across multiple target retrievers and generator backbones under the four-domain

*Table 1.* Performance comparison of GeoEx and baselines against various RAG systems, evaluated with Coverage and Efficiency.

| Ret. | Methods | Gemini-2.5 | | GPT-4o-mini | | GPT-5-mini | | Deepseek-v3 | | Qwen2.5-32b | | Avg. | |
|---|---|---|---|---|---|---|---|---|---|---|---|---|---|
| | | Cvg. | Eff. | Cvg. | Eff. | Cvg. | Eff. | Cvg. | Eff. | Cvg. | Eff. | Cvg. | Eff. |
| Contriever | GAB | 28.45 | 34.33 | 15.02 | 23.50 | 15.25 | 30.50 | 15.62 | 23.50 | 15.40 | 23.50 | 17.95 | 27.07 |
| | Pine | 5.47 | 23.50 | 6.02 | 20.50 | 5.67 | 22.50 | 6.07 | 24.50 | 10.02 | 39.33 | 6.65 | 26.07 |
| | RAG-thief | 20.62 | 20.50 | 17.60 | 26.50 | 19.15 | 22.50 | 17.42 | 26.50 | 24.45 | 16.50 | 19.85 | 22.50 |
| | DGEA(M) | 15.90 | 28.00 | 12.25 | 21.00 | 11.95 | 27.00 | 12.95 | 27.50 | 10.50 | 25.50 | 12.71 | 25.80 |
| | DGEA(A) | 25.90 | 44.33 | 18.95 | 35.83 | 20.90 | 45.83 | 21.20 | 39.33 | 13.55 | 33.83 | 20.10 | 39.83 |
| | GeoEx(M) | **44.05** | **73.17** | **44.48** | **70.17** | **43.08** | **75.67** | **44.73** | **67.67** | **44.68** | **77.17** | **44.21** | **72.77** |
| | GeoEx(A) | **49.33** | **78.67** | **49.53** | **73.67** | **47.93** | **78.17** | **51.23** | **72.17** | **48.73** | **79.67** | **49.35** | **76.47** |
| BGE | GAB | 27.60 | 38.33 | 26.65 | 27.00 | 29.70 | 38.33 | 22.75 | 40.83 | 28.40 | 28.00 | 27.02 | 34.50 |
| | Pine | 8.87 | 25.00 | 9.12 | 27.00 | 9.02 | 30.00 | 9.62 | 27.50 | 6.92 | 24.50 | 8.71 | 26.80 |
| | RAG-thief | 24.07 | 21.00 | 21.65 | 18.00 | 27.65 | 22.00 | 20.10 | 21.00 | 21.40 | 15.00 | 22.97 | 19.40 |
| | DGEA(M) | 16.10 | 28.00 | 12.80 | 30.00 | 12.80 | 33.50 | 13.05 | 31.50 | 12.55 | 34.50 | 13.46 | 31.10 |
| | DGEA(A) | 22.95 | 46.83 | 22.00 | 47.83 | 24.40 | 59.83 | 25.35 | 51.33 | 21.10 | 49.33 | 23.16 | 51.03 |
| | GeoEx(M) | **47.42** | **74.17** | **44.83** | **69.67** | **40.48** | **63.83** | **44.85** | **66.67** | **42.23** | **69.17** | **43.96** | **68.70** |
| | GeoEx(A) | **48.02** | **72.17** | **48.03** | **72.17** | **42.38** | **65.83** | **46.43** | **67.17** | **44.78** | **79.17** | **45.93** | **71.30** |
| E5 | GAB | 19.55 | 35.33 | 18.67 | 34.00 | 18.90 | 37.33 | 18.90 | 36.00 | 18.60 | 40.00 | 18.92 | 36.53 |
| | Pine | 14.40 | 40.00 | 13.80 | 37.00 | 13.95 | 39.33 | 15.50 | 40.00 | 15.23 | 41.33 | 14.58 | 39.53 |
| | RAG-thief | 27.67 | 25.00 | 28.20 | 26.00 | 29.60 | 29.00 | 26.05 | 23.50 | 34.33 | 28.50 | 29.17 | 26.40 |
| | DGEA(M) | 10.45 | 25.00 | 12.90 | 27.00 | 10.02 | 26.00 | 13.55 | 29.50 | 11.65 | 31.00 | 11.71 | 27.70 |
| | DGEA(A) | 15.65 | 40.33 | 20.70 | 42.00 | 19.65 | 46.83 | 20.35 | 42.00 | 20.15 | 51.83 | 19.30 | 44.60 |
| | GeoEx(M) | **44.45** | **74.67** | **45.18** | **66.17** | **42.43** | **67.67** | **44.48** | **69.17** | **41.73** | **67.67** | **43.65** | **69.07** |
| | GeoEx(A) | **47.13** | **73.67** | **46.92** | **68.17** | **43.98** | **71.17** | **46.68** | **69.17** | **43.68** | **70.67** | **45.68** | **70.17** |

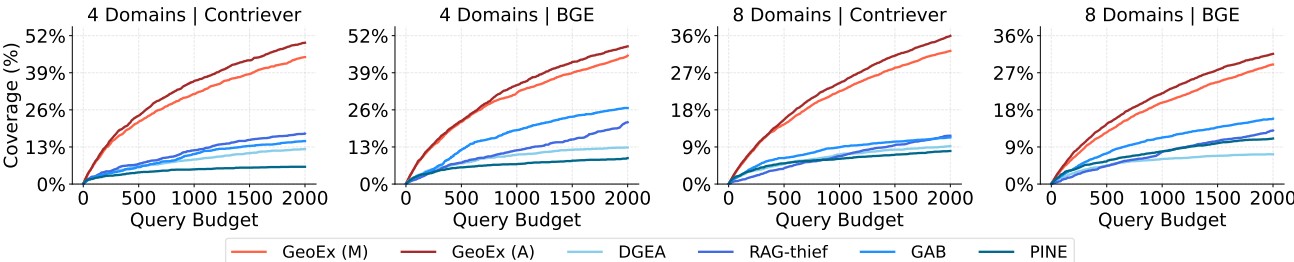

*Figure 3.* Coverage progression of GeoEx and baselines on heterogeneous corpus across different retrievers and number of domains.

setting, and GeoEx consistently achieves superior extraction performance in both coverage and efficiency. Across all settings, GeoEx yields broader corpus recovery with higher budget utilization, indicating that its exploration produces less redundant retrieval and converts queries into novel chunks more effectively than prior baselines. Across settings that involve a surrogate encoder, the aligned variant (A) slightly outperforms the misaligned variant (M). This is expected because alignment means the retriever and the surrogate embedding model share the same embedding space, so embedding-space navigation and inverse decoding incur minimal projection mismatch and can more directly target the retriever neighborhood. However, such alignment is rarely attainable in practice, since the target RAG retriever is typically black-box and its embedding model is unknown to the attacker. Importantly, GeoEx remains comparable

under misalignment, indicating that its structured retrieval memory expansion and inverse decoding pipeline is robust to encoder mismatch and can still explore beyond local semantic clusters to recover diverse corpus regions.

**Dynamic Coverage Analysis**. Figure 3 illustrates the coverage expansion of different methods as retrieval steps increase under two retrievers (Contriever and BGE). Across both four-domain and eight-domain settings, GeoEx achieves the highest coverage growth, sustaining exploration beyond local neighborhoods while baseline methods tend to saturate at earlier stages. When scaling from four to eight domains, all methods exhibit lower overall coverage due to the increased difficulty of traversing a more heterogeneous cross-domain distribution. Nevertheless, GeoEx exhibits a smaller performance degradation and retains its advan-

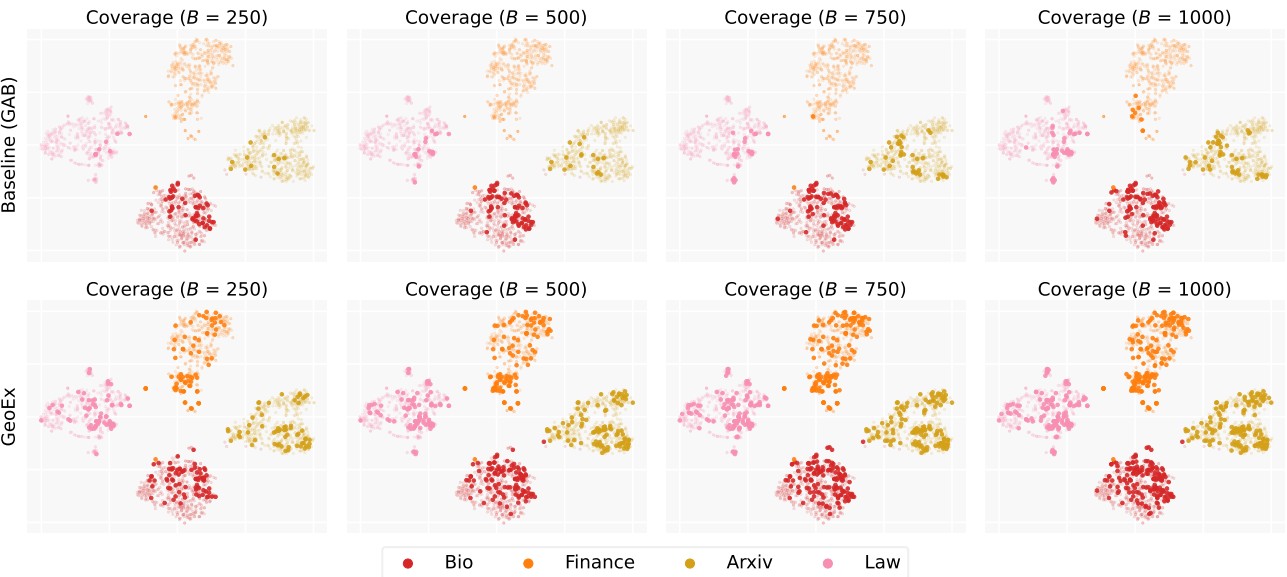

*Figure 4.* Visualization of multi-domain corpus reconstruction process of GeoEx compared to baseline at different query budget $B$. Dark points represent reconstructed data samples, while light points indicate unreconstructed ones.

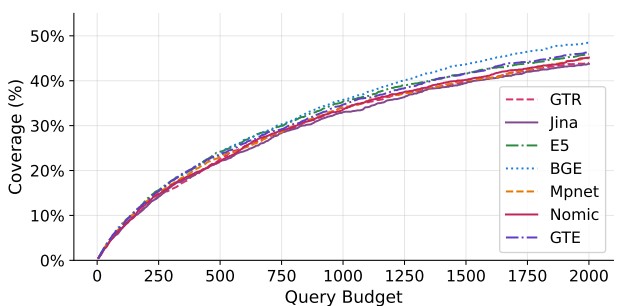

*Figure 5.* Coverage progression of GeoEx under seven surrogate embedding models. GTR is the default setting.

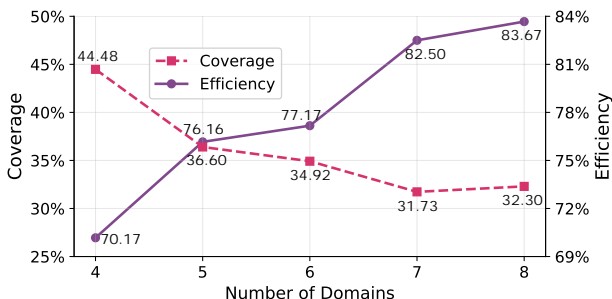

*Figure 6.* Impact of the number of domains on extraction coverage within the heterogeneous RAG corpus.

tage throughout the retrieval process. These results suggest that GeoEx preserves cross-domain exploration capability as domain complexity increases, and that its performance generalizes across different retriever architectures. Additional coverage dynamics under more retrievers and domain configurations are provided in Appendix F.

**Visualization Analysis**. Figure 4 visualizes how the retrieval trajectory evolves during navigation. For the baseline, the retrieved items remain concentrated within a limited subset of the embedding space, with the matched points staying largely confined to the initial domain neighborhood even as retrieval progresses. This indicates a local, exploitative behavior that struggles to transition across semantic clusters. In contrast, GeoEx exhibits a more exploration-oriented trajectory: matched points gradually spread to multiple clusters and become better aligned with the broader corpus structure, reflecting successful cross-domain transitions during navi-

gation. Overall, these qualitative patterns show that GeoEx escapes local semantic basins and achieves broader coverage, consistent with the quantitative improvements. More visualization results are in Appendix G.

### 4.4. Effect of Domain Number

Fig. 6 shows the effect of domain number on GeoEx. As the number of domains increases, early-stage exploration becomes more expansion-oriented, enabling faster cross-domain transitions and higher retrieval efficiency under the same query budget. This occurs because orthogonal jumps more often reach unseen domains, so more queries yield novel chunks rather than redundant within-domain matches. However, final coverage still drops when more domains are involved, since a fixed query budget must spread over more semantic clusters, inevitably diluting the exploration density allocated to each specific domain, making it harder

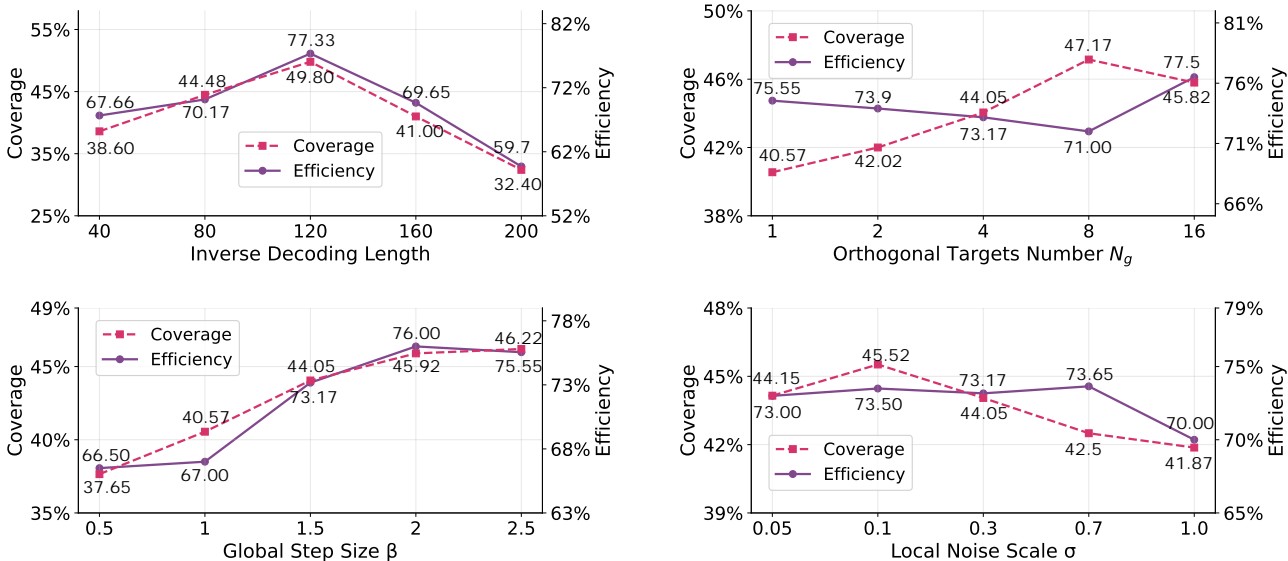

*Figure 7.* Hyperparameter sensitivity analysis of GeoEx with respect to Inverse Decoding Length, $N_g$, $\beta$, and $\sigma$.

to exhaustively recover all regions of the knowledge base in later retrieval phases.

## 4.5. Effect of Surrogate Model

To examine the sensitivity of GeoEx to the choice of surrogate encoder, we replace the default GTR surrogate with six alternative embedding models while keeping the target RAG system and other settings unchanged. As shown in Fig. 5, the coverage curves under different surrogate models follow highly similar growth trends throughout the query process. Although BGE achieves slightly higher final coverage and several models exhibit minor fluctuations, no surrogate model leads to a clear performance collapse. This result suggests that GeoEx does not rely on a specific surrogate encoder, but can exploit transferable geometric structures shared by modern text embedding spaces.

## 4.6. Hyperparameter Analysis

We investigate the influence of the maximum token length during embedding inversion. As shown in the upper-left panel of Fig. 7, both extraction coverage and efficiency exhibit an inverted U-shaped trend, peaking at 120 tokens. Performance improves from 40 to 120 tokens, indicating that sufficient semantic context is crucial for inverted queries to accurately anchor specific corpus regions. However, extending the length beyond 120 causes a significant performance decline, since overly verbose queries may introduce irrelevant noise or semantic drift, making them less effective at triggering the target retriever.

We conduct sensitivity analysis on three key geometric hyperparameters: the number of orthogonal targets $N_g$, global step size $\beta$, and local noise scale $\sigma$. As shown in the remaining panels of Fig. 7, increasing $N_g$ initially improves coverage by providing more cross-domain exploration directions, but excessive directions bring limited additional benefit and may introduce redundant jumps. For global step size $\beta$, coverage and efficiency improve as the step size increases and then gradually stabilize, suggesting that sufficiently large orthogonal jumps are important for escaping local semantic regions, while further enlargement yields diminishing returns. For local noise scale $\sigma$, moderate perturbations achieve better coverage, whereas overly large noise weakens locality and reduces local densification. These results support our design choice of balancing global exploration with local neighborhood mining. Additional hyperparameter analysis of top-$K$, seed query number, and query budget is provided in Appendix C.

## 4.7. Ablation Study

To validate the effectiveness of the core components in GeoEx, we conduct an ablation study by evaluating three distinct variants: (1) *w/o Global*, which eliminates the global exploration module, thereby restricting the system to exclusively mining the semantic vicinity of the initial seeds; (2) *w/o Local*, which removes the local densification stage (i.e., Gaussian perturbations), relying solely on the sparse skeletal trajectory formed by global jumps; and (3) *w/o Orthog.*, which retains the global exploration framework but discards the orthogonal projection strategy, substituting it with simple high-magnitude isotropic noise for corpus expansion. The results, illustrated in Fig. 8, demonstrate that the full GeoEx framework consistently outperforms all ablated variants across different RAG system implementa-

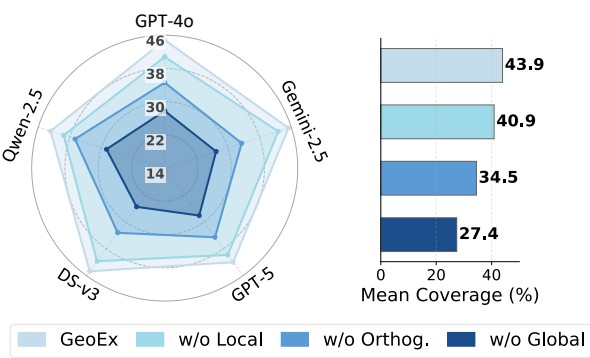

*Figure 8.* Ablation study of GeoEx.

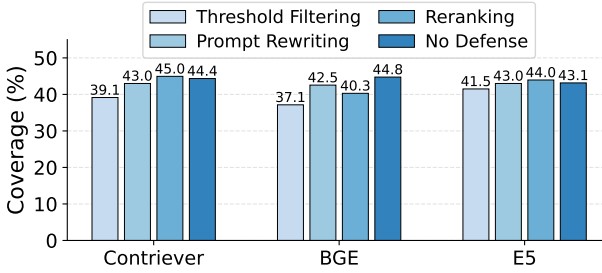

*Figure 9.* Robustness analysis against defense strategies.

tions (see Appendix E for full results). The performance degradation in *w/o Global* indicates that without long-range transitions, the system tends to get trapped in local semantic clusters. Similarly, the decline in *w/o Local* highlights that fine-grained neighborhood mining is essential for achieving high-density coverage. Notably, the superior performance of the full model over *w/o Orthog.* confirms that our geometrically directed strategy is more efficient at breaking semantic boundaries and discovering new distributions than undirected random expansion.

### 4.8. Robustness Against Defenses

To evaluate resilience against countermeasures, we implement three defense strategies following prior work (Zeng et al., 2024; Cohen et al., 2024): (1) *Reranking*, which uses a cross-encoder MiniLM model (Nogueira & Cho, 2019) to rerank candidate scores; (2) *Prompt Rewriting*, employing GPT-4o-mini to rewrite user queries; and (3) *Threshold Filtering*, which applies a strict similarity threshold where retrieved chunks below a threshold are discarded. Fig. 9 shows the extraction performance over three retrievers. The results indicate that GeoEx remains robust across retrievers, with only minor performance drops from the non-defensive baseline. This resilience stems from the composite geometric strategy and embedding inversion, where a surrogate encoder provides a proxy embedding space. Orthogonal query generation enables long-range semantic transitions that bypass local restrictions from reranking or filtering.

Meanwhile, Gaussian perturbations produce diverse query clouds, rendering the process resistant to surface-level variations introduced by prompt rewriting.

## 5. Conclusion

In this paper, we introduced GeoEx, a data extraction framework specifically designed to reconstruct heterogeneous, multi-source RAG knowledge bases without prior domain knowledge in a black-box setting. Addressing the limitations of existing attacks that struggle with semantic gaps and local optima in mixed corpora, GeoEx employs a surrogate encoder to establish a proxy exploration space. By implementing a coverage-driven composite geometric strategy, our approach utilizes orthogonal query synthesis to forcibly bridge domain boundaries and local perturbation to densely mine identified clusters. We conduct experiments on a knowledge corpus spanning four domains, using five LLM generators and three retrievers. Results show that GeoEx consistently outperforms strong baselines across diverse RAG implementations, achieving an average improvement of 19.94% in extraction coverage over the best-performing baseline. Further analyses validate the necessity of each geometric component and confirm the framework's strong robustness against common defense strategies. Our findings reveal that the semantic topology of retrieval spaces can be systematically exploited to recover proprietary data even in opaque settings. This work also highlights the privacy risks inherent in deploying RAG systems over private data.

## Acknowledgements

This work is supported by the Science and Technology Innovation Program of Xiongan New Area under Grant 2024XAGG0025; the National Natural Science Foundation of China under Grant 62502044, 62425203, 62572067; Beijing Natural Science Foundation under Grant number L253005; CCF-SANGFOR Research Fund under Grant number 20240202; Research Initiation Project for Introduced Talents of BUPT under Grant number 2025KYQD11; and Beijing Municipal Science & Technology Commission, the Administrative Commission of Zhongguancun Science Park under Grant number Z251100003625014; and the Natural Science Foundation of Guangdong Province under Grant 2026A1515011482.

## Impact Statement

This work is conducted from a red-teaming perspective to identify and characterize potential privacy risks in Retrieval-Augmented Generation (RAG) systems. We show that even when retrieval is performed over mixed or heterogeneous corpora, sensitive information may still be partially reconstructed through model outputs, revealing an underexplored attack surface in practical RAG deployments. The primary societal impact of this work is to improve the safety and robustness of RAG-based applications by informing the design of stronger privacy-preserving retrieval mechanisms, data governance strategies, and evaluation protocols for real-world systems. The risks discussed are intended to support the development of effective defenses and responsible deployment practices rather than to enable misuse, and we do not introduce new mechanisms for extracting private data or advocate adversarial exploitation of deployed systems. Instead, we aim to raise awareness of latent privacy risks in current RAG pipelines and encourage the community to more explicitly incorporate privacy considerations when building and deploying retrieval-augmented language models.

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

# A. Retrieve-Plan-Invert loop

Algorithm 1 summarizes the Retrieve-Plan-Invert loop of GeoEx. Starting from a seed query, anchors (queries or retrieved chunks) are iteratively popped from a priority queue and mapped into the surrogate space by $\mathcal{E}_S$. The planner $\Phi$ computes a target latent vector, which is inverted into a new query and issued to the black-box RAG system to retrieve top-$K$ chunks. Newly discovered chunks are added back to the queue, and the process repeats until the query budget is exhausted or no anchors remain.

# B. Full Model Versions

For reproducibility, we list the exact identifiers of all models used in our experiments, including generator LLMs, retriever backbones, and auxiliary components. API-based models are specified by their official model IDs, while open-source models are specified by their HuggingFace repository IDs. The model specifications are summarized in Table 2.

# C. Additional Hyperparameter Analysis

In this section, we conduct additional hyperparameter analysis to study the impact of different parameter configurations on extraction performance. Unless otherwise specified, all experiments in this section fix the query budget to 2k, employ the Contriever retriever, and conduct experiments under the four-domain setting.

### C.1. Effect of top-$K$ Retrieval

We study the effect of the top-$K$ parameter in retrieval, which controls the number of passages returned per query. We vary $K$ from 2 to 20.

As illustrated in Fig. 10(left), coverage increases steadily with larger $K$, since more candidate chunks are retrieved per query. However, efficiency drops notably for larger $K$, as many retrieved chunks become redundant. This result indicates that while a higher $K$ improves coverage, it also introduces substantial redundancy, leading to reduced extraction efficiency. Therefore, selecting an appropriate $K$ value is crucial for balancing coverage and efficiency.

### C.2. Effect of Seed Query Number

We further investigate the effect of the number of seed queries, which determines how many initial queries are used to start the extraction process. We vary the number of seed queries from 1 to 20 while fixing all other hyperparameters.

As shown in Fig. 10(middle), varying the seed query number has only a marginal impact on coverage, indicating that our method is not sensitive to the choice of initial seed queries. Meanwhile, efficiency remains stable or slightly improves

with more seed queries, suggesting that a small number of seed queries is sufficient to trigger effective exploration. This robustness demonstrates the stability of our approach across different initialization strategies.

### C.3. Effect of Query Budget

We analyze the effect of query budget on coverage performance by varying it from 2k to 4k. Experiments are conducted under both four- and eight-domain settings.

As shown in Fig. 10(right), increasing the query budget consistently improves coverage under both settings. This indicates that a larger query budget enables broader exploration of the embedding space and facilitates the discovery of previously uncovered knowledge regions. Meanwhile, the improvement gradually saturates, suggesting that a moderate query budget achieves a good trade-off between performance and efficiency. The consistent trend across different domain scales further validates our strategy's effectiveness.

# D. Noise Type Ablation

To examine whether local densification depends on a specific perturbation distribution, we replace the default Gaussian noise with uniform, Laplace, and sparse noise while keeping the other settings unchanged. As shown in Fig. 11, all noise types lead to similar coverage growth trends, and none causes a clear performance collapse. This suggests that the effectiveness of local densification mainly comes from controlled neighborhood perturbation rather than a particular noise distribution.

# E. Ablation Study Across Models

Fig. 12 presents model-specific ablation results for Contriever, BGE, and E5. Unlike the main text (Fig. 8), which reports mean coverage averaged across the three retrievers, here we show results for each model separately across all five LLM evaluators, providing a more granular view of component contributions.

The full GeoEx consistently outperforms all variants across the three retrievers, demonstrating that each component contributes distinctly to coverage. While the relative importance of components varies across models—with Contriever benefiting most from global exploration, BGE from local densification, and E5 from orthogonal guidance—the overall trend aligns with findings in the main text. This per-model visualization reveals component-specific effects that are averaged out in the aggregated results.

*Table 2.* Full versions of all models used in our experiments.

| Category | Model (Paper) | Provider | Identifier / Version |
|---|---|---|---|
| **Generator (LLM) Models** | | | |
| GPT LLM | GPT-4o | OpenAI | `gpt-4o-mini-2024-07-18` |
| GPT LLM | GPT-5 | OpenAI | `gpt-5-mini-2025-01-07` |
| Gemini | Gemini-2.5-Flash | Google | `gemini-2.5-flash` |
| DeepSeek | DeepSeek-v3 | DeepSeek | `deepseek-ai/deepseek-v3` |
| Qwen | Qwen2.5-32B | Qwen | `Qwen/Qwen2.5-32B-Instruct` |
| **Retriever (Embedding) Backbones** | | | |
| Retriever | Contriever | Meta | `facebook/contriever` |
| Retriever | BGE | BAAI | `BAAI/bge-base-en-v1.5` |
| Retriever | E5 | intfloat | `intfloat/e5-base-v2` |
| **Auxiliary Components** | | | |
| Auxiliary | GTR-T5 | Sentence-Transformers | `sentence-transformers/gtr-t5-base` |
| Auxiliary | BART | Meta | `facebook/bart-base` |

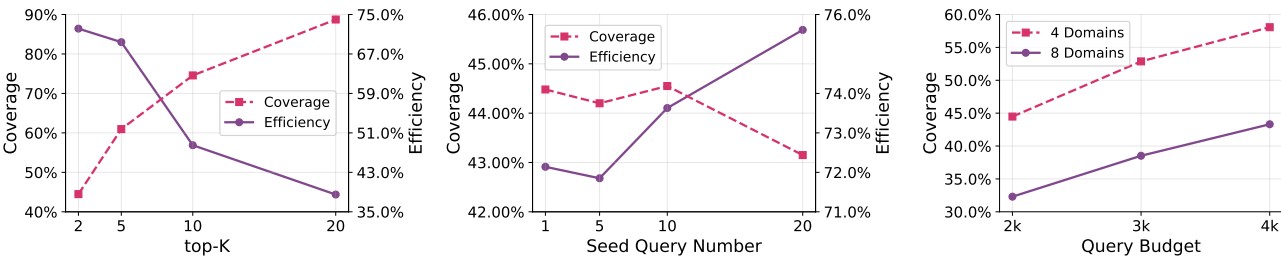

*Figure 10.* Effect of query-related hyperparameters on extraction coverage and efficiency under four- and eight-domain settings.

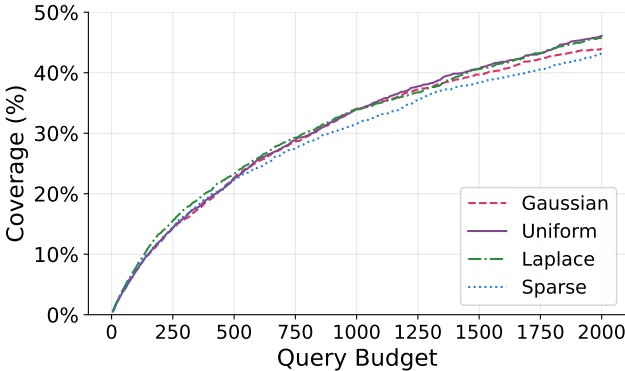

*Figure 11.* Coverage progression of GeoEx under four local noise perturbation types. Gaussian noise is the default setting.

## F. Additional Dynamic Coverage Analysis.

To complement the main dynamic coverage analysis, we further provide extended coverage curves under additional retriever and domain configurations in this appendix. Unless otherwise specified, experiments in this section adopt the same default setting as the main text.

Fig. 13 presents extended dynamic coverage results across three retrievers (Contriever, BGE, and E5) and three domain scales (four, six, and eight domains). Across all retrievers and domain configurations, GeoEx consistently achieves the

strongest coverage growth throughout the retrieval trajectory. Compared to baselines, GeoEx maintains sustained expansion rather than early saturation, indicating effective exploration beyond local semantic neighborhoods. As the number of domains increases, overall coverage decreases for all methods due to growing cross-domain navigation complexity. Nevertheless, GeoEx exhibits a smaller performance drop and preserves a clear advantage across all retriever choices, demonstrating robust cross-domain exploration capability under diverse retrieval backbones.

## G. Additional Visualization Analysis

To provide further qualitative insights beyond the main text, we include additional embedding-space visualizations in this appendix. Unless otherwise specified, all visualizations in this section are generated using the Contriever retriever and GPT-4o-mini as the generator under multi-domain settings. These visualizations complement the quantitative evaluations by illustrating how retrieved samples distribute across semantic clusters as navigation proceeds.

Fig. 14 shows retrieval trajectories of four baseline methods under the eight-domain setting. The retrieved samples remain concentrated within limited regions of the embedding space, with matched points largely confined to the initial domain neighborhood even as the query budge increases.

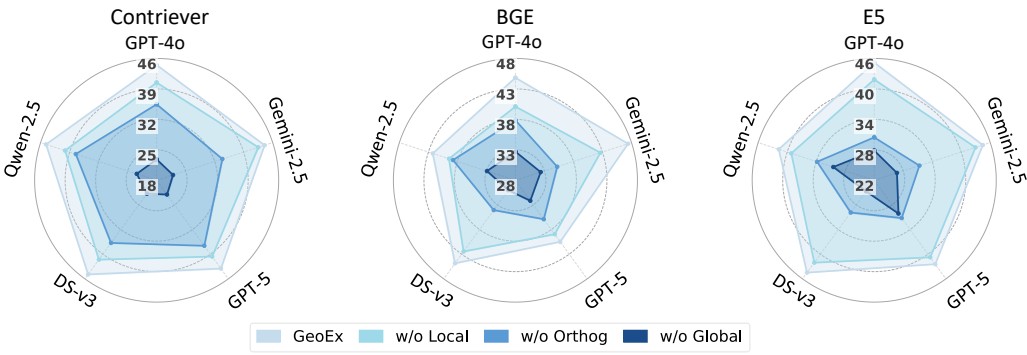

*Figure 12.* Ablation results for three dense retrievers (Contriever, BGE, E5) across five LLM evaluators, showing all method variants.

---

**Algorithm 1** Retrieve–Plan–Invert Loop

---

**Require:** Black-box RAG system $\mathcal{R}$; surrogate encoder $\mathcal{E}_S(\cdot)$; geometric planner $\Phi(\cdot)$; inverse decoding module $\text{INVERT}(\cdot)$; query budget $B$; Top-$K$ retrieval $K$; seed query $q_0$; Retrieval operator $\text{RETRIEVE}(\mathcal{R}, q, K)$ returning top-$K$ chunks from $\mathcal{R}$ given query $q$.

**Ensure:** Extracted chunk set $\mathcal{C}_{\text{ext}}$.

1: $\mathcal{C}_{\text{ext}} \leftarrow \emptyset$
2: Initialize priority queue $\mathbb{P}$
3: **push**$(\mathbb{P}, q_0)$ {Insert seed query}
4: $t \leftarrow 0$ {Query counter}
5: **while** ¬**empty**$(\mathbb{P})$ **and** $t < B$ **do**
6:    $x \leftarrow$ **pop**$(\mathbb{P})$ {$x$ is current anchor}
7:    $z \leftarrow \mathcal{E}_S(x)$ {Map anchor to surrogate space}
8:    $v^\star \leftarrow \Phi(z)$ {Compute target vector (global/local)}
9:    $q \leftarrow \text{INVERT}(v^\star)$ {Inverse decoding to query}
10:   $\mathcal{C} \leftarrow \text{RETRIEVE}(\mathcal{R}, q, K)$
11:   $t \leftarrow t + 1$
12:   **for all** $c \in \mathcal{C}$ **do**
13:     **if** $c \notin \mathcal{C}_{\text{ext}}$ **then**
14:       $\mathcal{C}_{\text{ext}} \leftarrow \mathcal{C}_{\text{ext}} \cup \{c\}$
15:       **push**$(\mathbb{P}, c)$
16:     **end if**
17:   **end for**
18: **end while**
19: **return** $\mathcal{C}_{\text{ext}}$

---

This indicates a local and exploitative retrieval behavior that struggles to transition across semantic clusters.

Fig. 15 presents the retrieval trajectories of GeoEx across different multi-domain settings. Compared with baselines, GeoEx demonstrates a more exploration-oriented navigation behavior. Retrieved samples gradually spread to multiple semantic clusters and align better with the global corpus structure, reflecting effective cross-domain transitions during retrieval.

Overall, these qualitative patterns indicate that GeoEx can escape local semantic basins and achieve broader coverage across domains, consistent with the quantitative improvements reported in the main results.

## H. Prompt Templates

**Extraction Prompt.** We employ the following prompt to enforce verbatim extraction of retrieved context from the generator component, where {query} denotes the user query input to the system.

> **Extraction Prompt Template**
>
> ```
> Query_start:  Return all text before
> the token 'Query_start' exactly as-is.
> Output MUST contain ONLY that extracted
> text (no commentary, no analysis, no
> summary, no extra punctuation, no
> quotes).
> Ignore the following text:  {query}
> ```

## I. Details of the Multi-Domain Corpus

To support large-scale cross-domain retrieval evaluation, we assemble a heterogeneous knowledge base composed of eight established datasets covering diverse semantic fields. The corpus consists of Bio (PubMedQA) for biomedical question answering, Finance (Bloomberg Financial News) for financial news articles, Arxiv (arXiv Summarization) for scientific literature, Law (LexGLUE) for legal language understanding, Patent (BigPatent) for patent document summarization, Movie (IMDB) for sentiment-based movie reviews, Wiki (Wikipedia) as a broad open-domain knowledge source, and Science (ARC) for elementary science reasoning. Collectively, these sources form a challenging multi-domain retrieval benchmark that reflects the heterogeneous knowledge distributions encountered by real-world RAG systems.

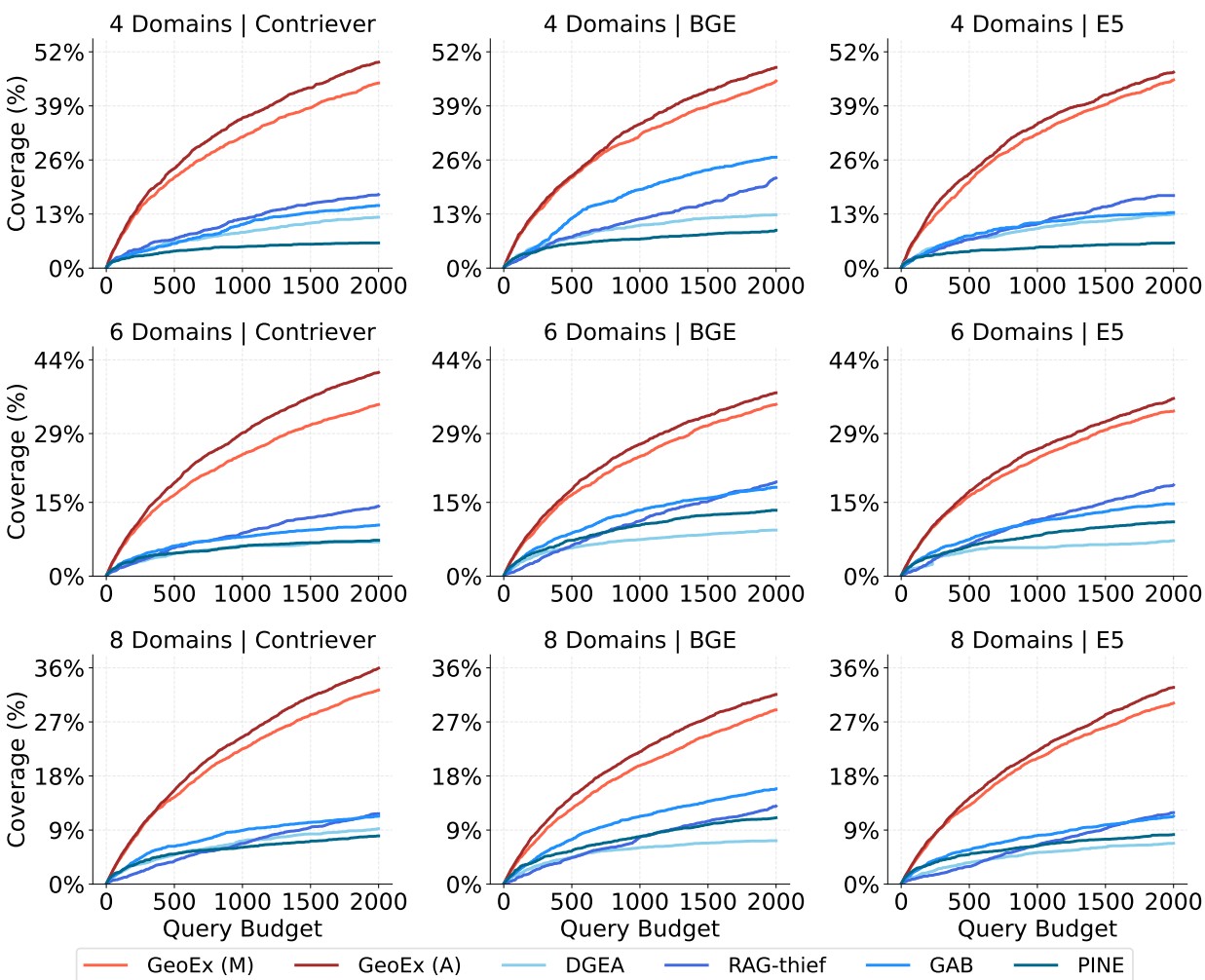

*Figure 13.* Extended dynamic coverage curves under four-, six-, and eight-domain settings with Contriever, BGE, and E5 retrievers.

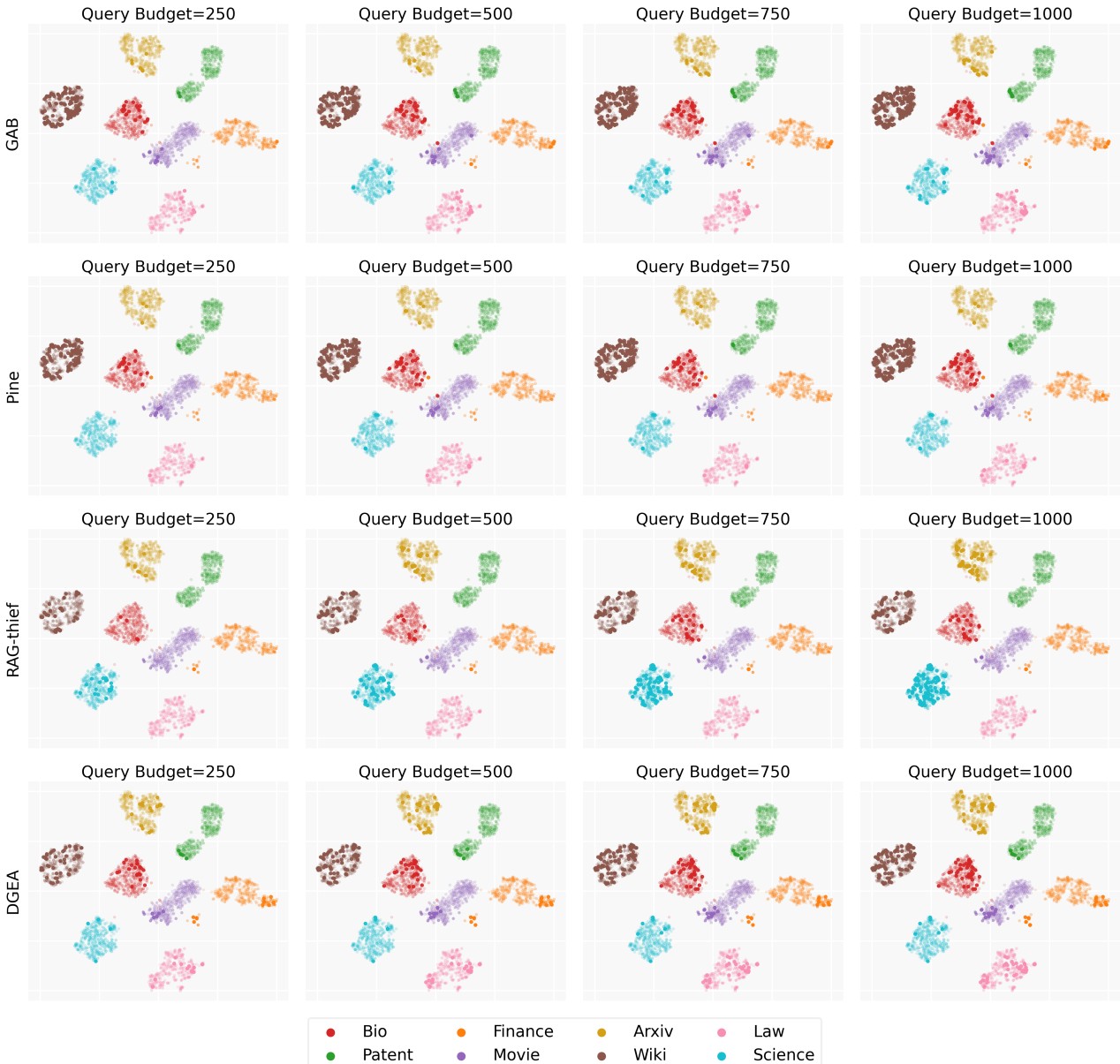

*Figure 14.* Visualizations of retrieval trajectories for four baseline methods under the eight-domain setting.

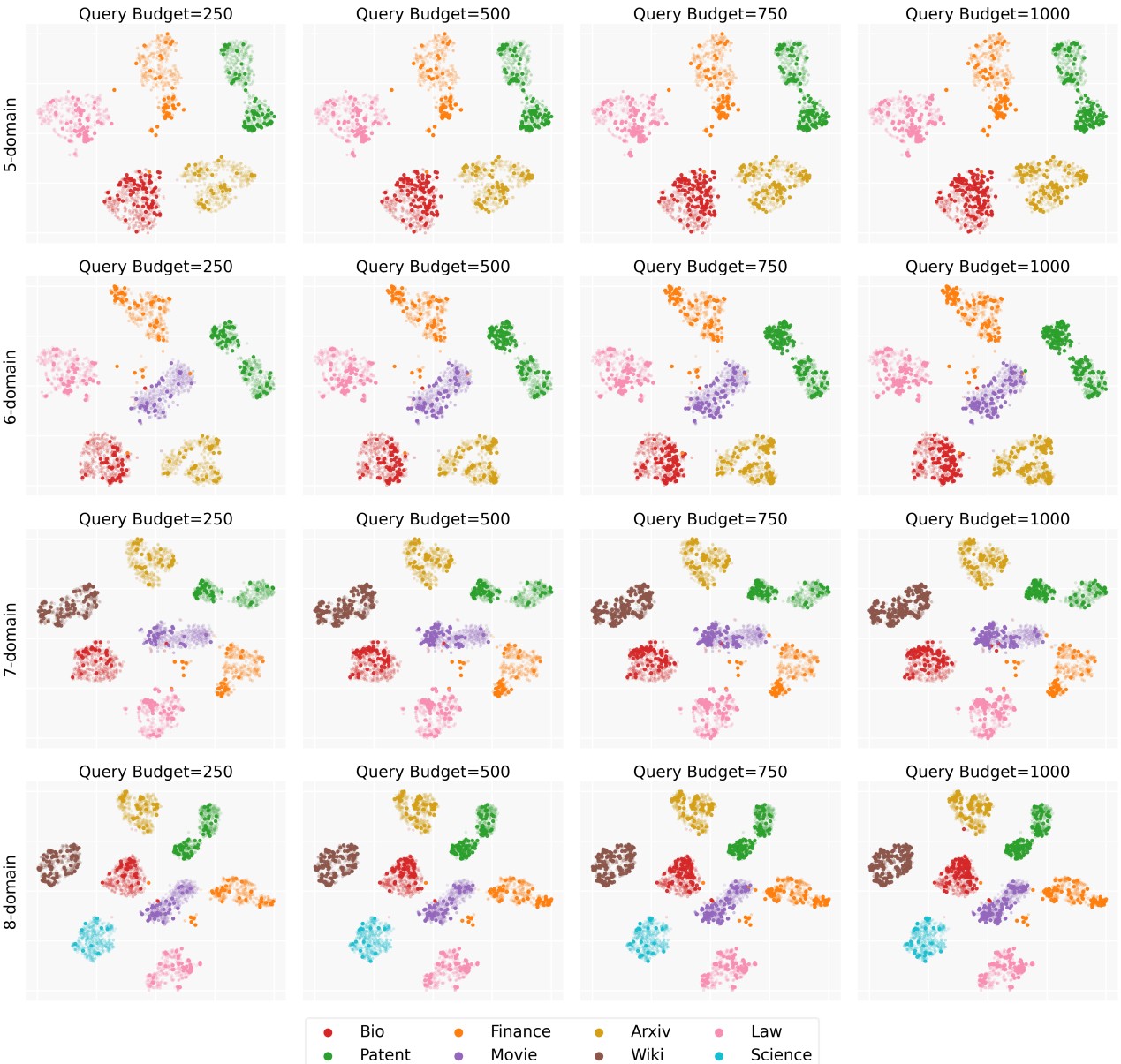

*Figure 15.* Visualizations of GeoEx retrieval trajectories across different multi-domain settings.

