# OpenReview forum: "Towards Whole-corpus Reconstruction of Heterogeneous RAG Knowledge Bases"
_ICML.cc/2026/Conference — ICML 2026 regular_

### Official Review · Reviewer_GXuV · 2026-03-10

**Soundness:** 2
**Presentation:** 3
**Significance:** 3
**Originality:** 3
**Overall Recommendation:** 4
**Confidence:** 3

**Summary:**

This paper proposes a novel extraction framework, GeoEx, designed to systematically reconstruct heterogeneous RAG knowledge bases under a no-prior setting. The framework adopts a Retrieve–Plan–Invert loop, where the generated queries are forwarded to the target RAG system for retrieval. By leveraging Global Exploration and Local Densification, GeoEx can efficiently navigate the heterogeneous corpus space. Experiments conducted on multiple types of datasets demonstrate that the proposed method achieves strong effectiveness.

**Compliance With Llm Reviewing Policy:**

Affirmed.

**Key Questions For Authors:**

1. Can the authors provide an additional case study or end-to-end example to better illustrate the full algorithmic workflow?

2. Can the authors further analyze the contribution of the Retrieve–Plan–Invert mechanism itself?

3. Have the authors considered alternative designs or replacements for the orthogonal module?

4. Can the authors provide a computational cost analysis of the proposed framework?

**Limitations:**

yes

**Strengths And Weaknesses:**

Strengths

1. This paper proposes a novel method for reconstructing heterogeneous RAG knowledge bases under a no-prior setting.

2. The experimental results demonstrate the effectiveness of the proposed method.

3. The paper is written in a relatively clear and understandable manner.


Weaknesses

1. It would be helpful to add an additional case study to facilitate understanding of the overall algorithmic workflow.

2. The ablation study validates the removal of the global and local components; it would be valuable to additionally assess how the Retrieve–Plan–Invert mechanism itself impacts overall performance.

3. Since the orthogonal module leads to the largest performance drop in the ablations, further analysis and experiments with alternative replacements for the orthogonal strategy would strengthen the paper.

4. Given that the algorithm involves relatively complex operations such as re-encoding and re-ranking, it would be helpful if the paper could provide some analysis of the computational cost or runtime overhead.

---

> ### Author Rebuttal · Authors · 2026-03-31
>
> We sincerely thank the reviewer for the constructive feedback, and the overall positive evaluation of our work. Below, we address your comments point by point:
>
> ### **W1: Case Study**
>
> Following Sec. 3, we illustrate one concrete iteration of GeoEx.
> Starting from a retrieved chunk $c$ (e.g., a biomedical passage on SNP and breast cancer survival, belonging to the *Bio* domain):
> ```
> A meta-analysis identified a common SNP rs8113308 ... indicating a predictive, treatment-specific effect on breast cancer survival.
> ```
> GeoEx first map the retrieved chunk $c$ into the surrogate space as $z_c = E_S(c)$.
> For global exploration, the planner constructs multiple orthogonal targets ${v^{*}_{\mathrm{global},i}} $, each of which can be inversely decoded into a query for cross-domain exploration.
> Each query retrieves cross-domain chunks, in this case, chunks belonging to the *Law* domain:
>
> ```
> Any notice, consent or other communication under this Agreement shall be in writing and shall be delivered personally...
> ```
> ```
> The Company (after giving effect to the transactions contemplated by this Agreement) is solvent...
> ```
>
> For local densification, GeoEx similarly generates multiple nearby targets ${v^{*}_{\mathrm{local},i}} $, which are decoded into queries used to retrieve semantically related biomedical evidence from the same *Bio* domain:
>
> ```
> PAI-1 mRNA was localized in cryostat thin eye sections via in situ hybridization analysis using specific 35S-labeled riboprobes...
> ```
> ```
> Axial stiffness and maximal load were inversely associated with circulating adiponectin levels ...
> ```
>
> This iterative retrieve-plan-invert loop progressively expands coverage across domains.
> Once these newly retrieved chunks are obtained, they are added into the query queue, encoded into the surrogate space, and then used in subsequent iterations to construct new global or local targets.
> A full case study will be updated to the manuscript.
>
>
> ### **W2: Mechanism Ablation**
> Thank you for this insightful comment.
> Following this suggestion, we add an ablation analysis on the overall Retrieve-Plan-Invert mechanism.
> Here, *Retrieve* is part of the underlying RAG system itself, while *Plan* is instantiated by our global exploration and local densification modules, which have already been analyzed in the existing ablation study (Fig.7).
>
> Therefore, we further ablate the *Invert* mechanism. Instead of projecting the planned latent targets back into executable text queries, we directly perturb the query in the discrete text space.
> Both the original results and the new ablation consistently show that **the overall inversion mechanism plays an important role**, as replacing it with direct text-space rewriting substantially weakens extraction effectiveness.
>
> | Method| Coverage | Efficiency |
> | - | -: | -: |
> | GeoEx|44.05 | 73.17 |
> | w/o Invert | 35.40 |59.50 |
>
> ### **W3: Orthogonal Module Alternatives**
> To explore alternative designs for cross-domain exploration, we further evaluate a replacement for the orthogonal direction construction using a *repulsive direction synthesis* method.
> Rather than generating strictly orthogonal directions, this strategy optimizes a set of diverse exploration directions around the current query embedding by encouraging them to spread out on the unit sphere while staying away from the current context vector.
>
> The results show that this replacement is also effective, achieving performance close to that of the original orthogonal design.
> This indicates that the benefit of our framework **does not rely on a single highly specific implementation**, while the slightly stronger performance of the original orthogonal module further supports our default design choice.
>
> |Method|Coverage|Efficiency|
> |-|-:|-:|
> | Orthogonal | 44.05 | 73.17 |
> | Alternative | 42.58 | 74.50 |
>
> ### **W4: Latency**
>
> We further analyze the runtime overhead of our method by reporting the **average time per newly recovered chunk** and comparing it against baselines.
> For fair efficency comparision, we deploy the proxy models used in our method and other baselines on a single H20 GPU.
> The results show that our method requires **the least time per newly recovered chunk** among all compared methods, outperforming baselines.  Overall, the results indicate that our method achieves strong extraction performance while maintaining good efficiency in practice.
> | Method    | Time   |
> | --------- | ------ |
> | RAG-Thief | 20.51s |
> | DGEA      | 36.87s |
> | GeoEx     | 9.06s  |
>
> We again thank the reviewer for the time and effort in helping improve our manuscript.
> We will incorporate these additional results and discussions in the revised version, and hope they help address your concerns.

---

> > ### Author Rebuttal · Reviewer_GXuV · 2026-04-03
> >
> > The rebuttal solved most of my concerns, but based on the overall quality of the paper, I will remain the original score of weak accept.

---

> > > ### Author Response · Authors · 2026-04-08
> > >
> > > Thank you for your time and for your encouraging feedback on our rebuttal. We are pleased to know that our response has addressed most of your concerns. We are grateful for your constructive suggestions, and we will incorporate them carefully in our revision.

---

### Official Review · Reviewer_tCWy · 2026-03-11

**Soundness:** 4
**Presentation:** 4
**Significance:** 3
**Originality:** 3
**Overall Recommendation:** 4
**Confidence:** 3

**Summary:**

This paper studies how to reconstruct the knowledge base of a black-box RAG system through query interactions. The authors propose GeoEx, which explores the retriever embedding space using global exploration to discover new semantic regions and local densification to extract more documents within a cluster. Experiments on heterogeneous multi-domain datasets show that GeoEx achieves higher corpus coverage and better efficiency than existing baselines.

The paper is clearly written and easy to follow, and the motivation and main idea are straightforward. The proposed method is simple but effective, and GeoEx consistently achieves better coverage and efficiency curves compared with the baselines. The experiments are also fairly comprehensive across different retrievers and settings.

One limitation is that the threat model and evaluation setup appear somewhat idealized. The experiments mainly evaluate coverage under a fixed query budget in a controlled benchmark, and it is unclear how well the attack would transfer to real-world RAG systems with stronger access restrictions or rate limits.

**Compliance With Llm Reviewing Policy:**

Affirmed.

**Key Questions For Authors:**

1. In the global exploration step, the paper constructs orthogonal directions in the embedding space to move toward new semantic regions. How sensitive is this design to the quality and geometry of the surrogate embedding model?
2. How does other noise work differently from Gaussian perturbation?

**Limitations:**

yes

**Strengths And Weaknesses:**

**Strenghs**
1. The paper is clearly written and easy to follow. The motivation, threat model, and the main idea of GeoEx are presented in a straightforward way, so it is easy to understand the core contribution of the paper.

2. The proposed method is simple but effective. The combination of global exploration and local densification is intuitive, and compared with the baselines, GeoEx shows consistently better coverage and efficiency curves across different settings.

3. The experiments are fairly comprehensive. The paper evaluates multiple retrievers and generator backbones, studies heterogeneous multi-domain settings, and includes additional analysis such as domain number, hyperparameters, and defense robustness. The task itself is also interesting and important, especially given the growing deployment of RAG systems.

**Weakness**
1. The threat model and evaluation setting may still be somewhat idealized. In particular, the paper mainly measures coverage with a fixed query budget in a controlled benchmark, but it is less clear how well the proposed attack would transfer to more realistic production RAG systems with stronger access restrictions, rate limits, retrieval filtering, or noisier generation behavior.

---

> ### Author Rebuttal · Authors · 2026-03-31
>
> We sincerely thank the reviewer for their helpful feedback, and the overall positive evaluation of our work. **All figure pdfs referenced below are available at** https://anonymous.4open.science/r/GeoEx_rebuttal-B05E.
> The tables here report the final coverage rates, while the figures additionally show the coverage progression as the number of queries increases.
>
> ### **W1: Real-world Restrictions**
> Thank you for this insightful comment.
> Our evaluation assumes a strict black-box setting with no prior knowledge of the corpus, retriever, or generator.
> We further simulate practical constraints by strictly limiting the retrieval window (top-K=2) and demonstrating robustness against defenses like retrieval filtering.
> Nevertheless, we acknowledge that our setting can be further improved to become more challenging:
>
> - Stronger access restrictions: Our main experiments already use a relatively restrictive setting with top-K=2.
> We further add a stricter top-K=1 setting, where **GeoEx still achieves comparable coverage and consistently outperforms baselines**:
>     |Method|Top-k=1|Top-k=2|
>     |-|-:|-:|
>     |GAB|12.32|28.45|
>     |Pine|4.62|5.47|
>     |RAG-thief|10.32|20.62|
>     |DGEA|10.17|15.9|
>     |GeoEx|34.45|44.05|
>
> - Retrieval filtering: We have already included robustness experiments with retrieval reranking and filtering (discarding chunks below a specific cutoff to block weakly relevant content) in Fig. 8.
> The results show GeoEx remains effective under these stronger retrieval controls.
>
> - Noisier generation behavior: We further add experiments with increased decoding temperature to simulate noisier generation (fig in `temperature.pdf`). The results show that **GeoEx remains robust under higher generation noise**.
>     |Temperature|Coverage|Efficiency|
>     |-|-:|-:|
>     |0.2|44.17|73.67|
>     |0.5|43.65|72.50|
>     |0.8|44.05|73.17|
>     |1.0|43.52|77.17|
>
> - Rate limits: This is a system-level constraint that would affect all extraction methods, which is largely orthogonal to the algorithmic design of the attack itself.
> In practice, such restrictions may also be alleviated through system-level strategies, such as using multiple accounts or increasing parallel querying capacity.
>
> We will incorporate these additional results and discussion in the revised version.
>
> ### **Q1: Surrogate Model Sensitivity**
> We deeply appreciate this insightful question.
> In the main experiments, we use a single surrogate model while transferring to multiple heterogeneous retrievers, and the strong performance across different retrievers shows that the GeoEx does not depend on a specific surrogate-target pairing.
>
> To further examine this point, we add new experiments by fixing the target retriever and varying the surrogate embedding model across several alternatives with different architectures and representation characteristics (fig in `surrogate.pdf`).
>
> The results demonstrate that **GeoEx maintains strong extraction coverage across all tested surrogate models**, confirming that our orthogonal jump strategy is highly generalizable.
> As long as the surrogate model possesses a reasonable semantic representation of language, the fundamental geometric properties of high-dimensional clustering hold true, allowing our framework to effectively bridge semantic gaps.
> |Surrogate Model|Coverage|Efficiency|
> |-|-:|-:|
> |gtr-t5-base (default)|44.05|73.17|
> |jina-embeddings-v2-base-en|43.77|67.50|
> |e5-base-v2|45.92|74.67|
> |bge-base-en-v1.5|48.52|78.00|
> |all-mpnet-base-v2|45.02|75.17|
> |nomic-embed-text-v1.5|45.17|68.50|
> |gte-base-en-v1.5|46.47|74.50|
>
> ### **Q2: Noise Type Ablation**
> We thank the reviewer for this helpful question.
> To examine whether the local densification effect depends specifically on Gaussian perturbation, we add experiments with several alternative noise types, including uniform noise, Laplace noise, and sparse noise (fig in `noise.pdf`).
>
> Overall, all these noise types remain effective for local densification and lead to similar performance trends.
> This suggests that the key factor is not the exact noise, but the ability to introduce controlled local perturbations.
> |Noise Type|Coverage|Efficiency|
> |-|-:|-:|
> |Gaussian|44.05|73.17|
> |Uniform|46.10|73.67|
> |Laplace|45.77|75.67|
> |Sparse|43.27|77.50|
>
> We again thank the reviewer for the constructive suggestions.
> We will incorporate these additional results and discussions in the revised version, and hope they help address your concerns.

---

> > ### Author Rebuttal · Reviewer_tCWy · 2026-04-03
> >
> > The concern get fully resloved. I will maintain the score af weak accept.

---

> > > ### Author Response · Authors · 2026-04-08
> > >
> > > Thank you for your time and for your positive feedback on our rebuttal. We are glad that our response has addressed most of your concerns. We sincerely appreciate your support and valuable suggestions, and we will incorporate them in the revised version.

---

### Official Review · Reviewer_yQfH · 2026-03-13

**Soundness:** 2
**Presentation:** 3
**Significance:** 2
**Originality:** 2
**Overall Recommendation:** 3
**Confidence:** 4

**Summary:**

This paper studies whole-corpus extraction attacks against black-box RAG systems when the underlying knowledge base is heterogeneous across domains and the attacker has no prior knowledge of the corpus. The authors propose GeoEx, which plans exploration in a surrogate embedding space, then inverts planned latent vectors into executable text queries using a similarity-guided decoding procedure. This method combines orthogonal global jumps for cross-domain exploration with local Gaussian perturbations for within-cluster densification. The authors conduct empirical evaluations on mixed corpora from up to eight different domains, across several retrievers and LLM generators, report improved extraction coverage and query efficiency over prior attack baselines.

**Compliance With Llm Reviewing Policy:**

Affirmed.

**Final Justification:**

I'd like to keep my score due to the unclear method details and insufficient experiment.

**Key Questions For Authors:**

NA

**Strengths And Weaknesses:**

Strengths:
1. The paper addresses a realistic and important threat model. The no-prior, black-box, heterogeneous-corpus setting is more challenging and more realistic than targeted extraction with strong prior hints.
2. The main empirical results are strong and consistent. In Table 1, GeoEx beats all baselines by a wide margin across retrievers and generators. Even the misaligned variant remains much stronger than prior methods, which is important because the aligned setting is unrealistic in practice.
3. The figures are generally useful. Figure 2 clarifies the architecture and search loop. Figure 3 supports the claim that GeoEx sustains coverage growth longer than baselines. Figure 4 gives an intuitive picture of why the method may work better in heterogeneous spaces, namely that it reaches multiple clusters instead of circling one.
4. The paper is reasonably complete. It includes ablations, hyperparameter analysis, and a defense section, which is more than many attack papers provide.

Weaknesses:
1. In Section 3.2, the language around orthogonality is too confident. Calling orthogonal synthesis the “optimal geometric solution” is not justified by the paper. This is geometric intuition, not theory. This matters because the paper’s conceptual pitch rests heavily on that rationale. As written, the method is a plausible heuristic, not an established principle.
2. On novelty and technical soundness, the paper is a creative combination of known ingredients rather than a fundamentally new technical primitive. Surrogate embedding spaces, inversion of embeddings to text, and local/global search heuristics all exist in adjacent literature.
3. Coverage is defined over chunks, but the paper does not state in the main body how many chunks each domain contributes, how chunking is performed, whether chunks overlap, or whether domains are balanced. This is a serious omission because chunk granularity can dramatically change both coverage and efficiency. A method that retrieves many short chunks can look better than one retrieving fewer long chunks, even if the actual information recovered is similar.
4. the ablation in Figure 7 is useful, but still somewhat coarse. It averages over retrievers, and the main figure reports only mean coverage. Since the paper is about balancing global and local search, I would like to see a more explicit sensitivity analysis over the number of orthogonal targets, the global step size β, and the local noise scale σ in the main paper. Those are the parameters most tied to the paper’s claimed contribution.
5. Section 4.7 and Figure 8 suggest GeoEx is robust to reranking, prompt rewriting, and threshold filtering, but the defense settings are not described in enough detail in the main text. Without exact thresholds, rewriting prompts, and reranking protocol, the robustness claim is hard to assess. This matters because attack papers should be careful not to overclaim resilience based on a thin defense suite.
6. There are several language problems, for example “since the RAG retriever in unknown” on Page 2.

---

> ### Author Rebuttal · Authors · 2026-03-31
>
> We sincerely thank the reviewer for the constructive comments. **All figure pdfs referenced below are available at**: https://anonymous.4open.science/r/GeoEx_rebuttal-B05E.
>
> **W1**: Theoretical Analysis
>
> We agree that calling orthogonal synthesis the “optimal geometric solution” was too strong.
> Rather than claiming optimality, we demonstrate that concentration of measure naturally renders independent domains nearly orthogonal, making our approach a mathematically principled heuristic.
> Specifically, we model a heterogeneous corpus as a mixture of isotropic Gaussian clusters in the embedding space:
> $$
> x \sim \sum_{m=1}^{M} \pi_m \mathcal{N}(\mu_m,\sigma^2 I),
> $$
> where the domain centroids $\mu_m$ are drawn i.i.d. from a zero-mean isotropic prior distribution.
> In high-dimensional spaces, a classical concentration of measure result states that independent random vectors, such as two samples drawn from different domains under this model, are nearly orthogonal with high probability.
> By explicitly enforcing orthogonality, our framework structurally directs the search away from the current local region, which mathematically maximizes the probability of intersecting the distribution of a new, independent domain.
>
> **W2**: Novelty
>
> While building upon existing components, our core contribution is to address a fundamentally novel research problem: whole-corpus extraction of heterogeneous RAG systems in a no-prior, black-box setting.
> Existing extraction attacks rely on discrete textual heuristics, which inevitably fail on multi-source corpora by getting trapped in local semantic optima.
> To overcome this bottleneck, our methodological novelty lies in a composite geometric strategy that bridges domain boundaries, coupled with inverse decoding to translate these latent plans into executable queries.
>
> **W3**: Chunking Protocol
>
> In our experiments, all corpora are chunked into segments of 512 tokens with no overlap and are evenly balanced across domains.
> Except for the last chunk of each document, all other chunks are of a fixed size, ensuring overall consistent lengths.
> Therefore, our evaluation protocol provides a reliable metric to measure the actual amount of compromised data.
>
> To further verify that our conclusions do not depend on chunk-level counting alone, we additionally report a token-level coverage (**Cvg.**), which measures recovered content by token volume rather than by chunk count (see fig in `token_cvg.pdf`).
> The token-level results are **highly consistent with chunk-level**, suggesting that our gains are not an artifact of retrieving disproportionately many short chunks.
>
> |Method|Chunk Cvg.|Token Cvg.|
> |-|-:|-:|
> |GAB|28.45|21.23|
> |PINE|5.47|4.94|
> |Thief|20.62|16.49|
> |DGEA(M)|15.90|15.56|
> |DGEA(A)|25.90|23.22|
> |GeoEx(M)|44.05|44.94|
> |GeoEx(A)|49.33|50.47|
>
> **W4**: Sensitivity Analysis
>
> We agree that a more fine-grained ablation analysis is valuable.
> Due to space limit, we had to present the corresponding retriever-specific results in **Appendix Fig.10 of our original submission**.
>
> We further provide sensitivity results on these hyperparameters (fig in `hyper.pdf`).
> The results demonstrate our intended balance between global exploration and local densification:
> - $N_g$: Coverage first increases and then slightly declines. A moderate number of directions is sufficient for cross-domain exploration, while too many introduce redundant jumps.
> - $\beta$: Coverage improves and then gradually stabilizes. Sufficiently large steps are crucial for escaping local regions, but further enlargement brings limited additional benefit.
> - $\sigma$: Coverage first improves and then decreases. Small perturbations effectively explore nearby content, whereas overly large noise weakens locality.
>
> |Hyper|Val.|Cvg.|
> |-|-:|-:|
> |$N_g$|1|40.57|
> ||2|42.02|
> ||4|44.05|
> ||8|47.17|
> ||16|45.82|
> |$\beta$|0.5|37.65|
> ||1.0|40.57|
> ||1.5|44.05|
> ||2.0|45.92|
> ||2.5|46.22|
> |$\sigma$|0.05|44.15|
> ||0.1|45.52|
> ||0.3|44.05|
> ||0.7|42.50|
> ||1.0|41.87|
>
> **W5**: Defense Strategies
>
> - Prompt Rewriting: Uses `GPT-4o-mini` to rewrite queries, disrupting exact wordings to increase retrieval difficulty. A shortened version of the prompt is shown below:
> ```
> Given an input query, rewrite it to improve fluency and naturalness while preserving the original semantic intent.
> Requirements:
> - Keep the core semantic intent unchanged.
> - Do not add new information, constraints, or details.
> ...
> - Output only the rewritten query.
> Input query:
> {query}
> ```
> - Threshold Filtering: Discards retrieved chunks with similarity scores below a specific cutoff, increasing attack difficulty by blocking weakly relevant content. We use model-specific thresholds, empirically calibrated via 200 queries per retriever to establish a strict relevance lower bound.
> - Reranking: Expands the initial retrieval pool to top-10 candidates, then reranks them using `MiniLM-L-6-v2` to select the final top-2 chunks.
>
> We will incorporate these discussions and revisions, and hope they address your concerns.

---

> > ### Author Rebuttal · Reviewer_yQfH · 2026-04-03
> >
> > Thanks for your response. I'd like to keep my score due to the unclear method details and insufficient experiment. Specifically, I still have the following concerns.
> >
> > 1. The authors construct a multi-domain corpus by integrating eight existing datasets (PubMedQA, Bloomberg, arXiv, LexGLUE, BigPatent, IMDB, Wikipedia, ARC). However, the paper does not justify why these specific datasets were chosen or why they collectively constitute a suitable evaluation benchmark for heterogeneous RAG extraction. Moreover, while the authors position their evaluation as multi-domain, they do not compare against or adopt widely recognized multi-domain RAG benchmarks (e.g., UltraDomain[1], DRAGONBench[2]) that have gained community acceptance. The absence of standard benchmarks limits generalizability and external validity.
> >
> > 2. The evaluation metrics are ambiguous. In heterogeneous RAG systems, does the full chunk set refer to the union of all documents across all sub-corpora, the total number of passages after chunking, or the total number of unique sentences? The chunking strategy itself, which directly influences KB, is not standardized across datasets, making cross-domain comparisons unreliable. Furthermore, coverage is defined as the percentage of unique chunks retrieved, but this metric treats all chunks equally, regardless of their semantic or informational significance. A system that successfully extracts 100% of low-information or redundant chunks would achieve perfect coverage, while a system that extracts 50% of high-value chunks containing critical knowledge would be scored lower. Without semantic weighting or information-theoretic measures, coverage alone is a poor proxy for successful corpus reconstruction.
> >
> > 3.  The overall RAG performance evaluation is missing. The paper reports only extraction metrics (coverage/efficiency) but provides no end-to-end RAG performance assessment (e.g., answer accuracy, F1 score). It remains unclear whether corpus reconstruction actually improves practical RAG utility.
> >
> > 4. Lack of latest baselines. GeoEx is compared against only four baselines, omitting recent strong methods such as CopyBreakRAG[3], MARAGE[4], and Pirates of the RAG[5]. This weakens the claim of state-of-the-art performance.
> >
> > 5. The paper evaluates multiple LLMs but does not analyze how model size and model architecture affect extraction performance.
> >
> > [1] MemoRAG: Boosting Long Context Processing with Global Memory-Enhanced Retrieval Augmentation.
> >
> > [2] DRAGON: Domain-specific Robust Automatic Data Generation for RAG Optimization.
> >
> > [3] Feedback-Guided Extraction of Knowledge Base from Retrieval-Augmented LLM Applications.
> >
> > [4] MARAGE: Multi-Model Adversarial Attack for Retrieval-Augmented Generation Database Extraction.
> >
> > [5] Pirates of the RAG: Adaptively Attacking LLMs to Leak Knowledge Bases.

---

> > > ### Author Response · Authors · 2026-04-08
> > >
> > > Thank you for the valuable feedback and constructive suggestions.
> > > ## Corpus Construction
> > > As discussed in Sec. 1, real-world RAG knowledge bases are often multi-source and heterogeneous, while existing extraction settings are typically evaluated on more homogeneous corpora, therefore do not fully capture the stronger attack scenario we study.
> > >
> > > To better reflect this challenge, we construct a controlled multi-domain evaluation corpus for black-box, no-prior, whole-corpus extraction.
> > > Importantly, it is built from **widely used public datasets** from different domains, not a newly collected private dataset.
> > > This design also allows us to explicitly control the number and composition of domains in the mixed corpus, supporting our later evaluations under different heterogeneous settings.
> > > Overall, our evaluation remains fair, reproducible, and well aligned with the task studied here.
> > >
> > > We also followed the reviewer's suggestion and evaluated GeoEx on UltraDomain and DRAGONBench.
> > > The results show that **GeoEx substantially outperforms baselines on both new benchmarks**.
> > >
> > > **Table 1: Additional results on new corpus**
> > > |Corpus|Method|Cvg.|
> > > |-|-|-|
> > > |UltraDomain|RAG-thief|25.20|
> > > ||GAB|30.75|
> > > ||GeoEx|53.90|
> > > |DRAGONBench|RAG-thief|28.00|
> > > ||GAB|27.35|
> > > ||GeoEx|47.85|
> > >
> > > ## Eval Metrics
> > > As we clarified in our phase-1 rebuttal, all documents are chunked using the same protocol: 512 tokens per chunk, with no overlap.
> > > The full chunk set / KB refers to the union of all chunks obtained after applying a unified chunking protocol across domains.
> > > We have followed the reviewer's phase-1 suggestion to report token-level coverage, and the results remained consistent with our main findings.
> > >
> > > Since the reviewer has now raised a new concern regarding token-level coverage, we also followed this suggestion and further added **informativeness-level coverage**.
> > > Specifically, each token's informativeness is computed as the negative log of its frequency, giving higher scores to rarer tokens.
> > > We then measure how much of the knowledge base's total informativeness is covered by the extracted tokens.
> > > The results show that our method **consistently performs best at the chunk, token, and informativeness-level coverage**.
> > >
> > > **Table 2: Coverage under different metrics**
> > > |Method|Chunk Cvg.|Token Cvg.|Informativeness Cvg.|
> > > |-|-|-|-|
> > > |GAB|28.45|21.23|23.89|
> > > |Pine|5.47|4.94|4.79|
> > > |RAG-thief|20.62|16.49|21.76|
> > > |CopyBreakRAG|24.82|22.35|26.66|
> > > |Pirate|18.62|16.08|16.87|
> > > |DGEA|15.90|15.56|10.99|
> > > |GeoEx(M)|44.05|44.94|43.01|
> > > |GeoEx(A)|49.33|50.47|47.42|
> > >
> > > These results further strengthen the reliability of our current evaluation, and we will include this analysis in the revised manuscript.
> > >
> > > ## RAG Utility
> > > We would like to respectfully clarify a fundamental **misunderstanding regarding the core objective** of our work.
> > > Our paper studies an attack that extracts the underlying database of RAG systems, rather than a method to improve RAG utility.
> > > Accordingly, the standard evaluation setting in prior works[1–4] focuses on extraction performance (e.g., extraction coverage rate), not downstream QA utility.
> > >
> > > [1] The Good and The Bad: Exploring Privacy Issues
> > > in Retrieval-Augmented Generation. ACL 2024.
> > > [2] Follow My Instruction and Spill the Beans: Scalable Data Extraction from Retrieval-Augmented Generation Systems. ICLR 2025.
> > > [3] RAG-Thief: Scalable Extraction of Private Data from Retrieval-Augmented Generation Applications with Agent-based Attacks. arXiv 2024.
> > > [4] Pirates of the RAG: Adaptively Attacking LLMs to Leak Knowledge Bases. arXiv 2024.
> > >
> > > ## Baselines
> > > Our original baselines were chosen to cover representative attack paradigms, and the methods mentioned by the reviewer largely fall into the same categories as those already evaluated.
> > >
> > > Specifically, CopyBreakRAG and Pirates (already discussed in Sec. 2 as Di Maio et al., 2024) are both agent-based adaptive extraction methods driven by attack feedback, therefore closely aligned with RAG-Thief, which is already included as a baseline.
> > > We do not include MARAGE as a direct baseline because it mainly optimizes a jailbreak suffix to force verbatim leakage of retrieved content, rather than solving whole-corpus extraction planning, thus it is orthogonal to our method.
> > >
> > > We have now added results for CopyBreakRAG and Pirates of the RAG, and **GeoEx remains stronger**:
> > >
> > > **Table 3: Comparison with more baselines**
> > > |Method|Cvg.|Eff.|
> > > |-|-|-|
> > > |GAB|28.45|34.33|
> > > |Pine|5.47|23.50|
> > > |RAG-thief|20.62|20.50|
> > > |CopyBreakRAG|24.80|25.50|
> > > |Pirate|18.62|28.67|
> > > |DGEA|15.90|28.00|
> > > |GeoEx|44.05|73.17|
> > >
> > > ## LLM Architectures
> > > We evaluated our attack against representative SOTA LLMs (including GPT-5, Gemini, Qwen, and DeepSeek) in a realistic black-box setting.
> > > These LLMs differ substantially in scale, architecture, and model family, already providing strong evidence for the robustness and generalization of GeoEx.
> > > We therefore believe this evaluation is sufficiently representative for the practical attack scenario studied in the paper.

---

### Decision · Program_Chairs · 2026-04-30

**Decision:**

Accept (regular)

**Comment:**

- This paper studies the reconstruction of heterogeneous RAG knowledge bases under a no‑prior black‑box threat model and proposes GeoEx, a framework combining global exploration and local densification in embedding space. The problem is timely and relevant given the growing deployment of RAG systems and associated privacy risks.

- Two reviewers initially rated the paper as Weak Accept, highlighting the empirical effectiveness of the proposed retrieve–plan–invert framework across heterogeneous multi‑domain settings. While concerns were raised regarding evaluation realism, component contributions, robustness under stricter constraints, and computational overhead, the authors made a substantive effort during rebuttal to address these issues by providing additional ablations, sensitivity analyses, runtime evaluation, robustness experiments, and clarification of the evaluation protocol. These responses were acknowledged by both reviewers as adequately resolving their primary technical concerns.

- One reviewer maintained a Weak Reject recommendation, primarily citing broader concerns on benchmark selection and evaluation metrics. The authors responded by adding additional benchmark evaluations and alternative coverage metrics, which partially address these concerns, although some disagreement remains regarding evaluation scope.

- Overall, the remaining concerns appear to relate more to evaluation framing than to the technical soundness of the proposed method. Given the importance of the problem setting, consistent empirical improvements over prior approaches, and the authors’ constructive rebuttal that resolved most technical concerns raised by multiple reviewers, I recommend **Weak Accept**.